# TABFLEX: SCALING TABULAR LEARNING TO MILLIONS WITH LINEAR ATTENTION

## ABSTRACT

Recent advances in the field of in-context learning (ICL) have demonstrated impressive performance for tabular classification, exemplified by TABPFN's success on small datasets. However, the quadratic complexity of the attention mechanism limits its applicability to larger datasets. To address this issue, we conduct a comprehensive comparison of popular scalable attention alternatives, including state-space models (SSMs) and linear attention mechanisms, revealing that the inherent causality of SSMs hinders ICL performance for large datasets, while linear attention preserves effectiveness. Leveraging these insights, we introduce TABFLEX, a model based on linear attention that supports thousands of features and hundreds of classes, capable of handling datasets with millions of samples. Extensive experiments demonstrate that TABFLEX is significantly faster than most existing methods while achieving top-two performance on small datasets among 25 baselines, with a $2\times$ speedup over TABPFN and a $1.5\times$ speedup over XGBoost. On large datasets, TABFLEX remains efficient (e.g., approximately 5 seconds on the `poker-hand` dataset, which consists of millions of samples), while achieving relatively solid performance.

## 1 INTRODUCTION

In recent years, Large language Models (LLMs) have achieved breakthroughs not only in language tasks (Achiam et al., 2023; Brown et al., 2020; Bai et al., 2023a; Dubey et al., 2024; Gemini Team et al., 2023) but also in handling diverse data modalities, including vision (Bai et al., 2023b; Gemini Team et al., 2023) and audio (Chu et al., 2023; 2024; Gemini Team et al., 2023). Their success stems from the underlying transformer architecture, which uses attention mechanisms (Vaswani et al., 2017) to capture complex patterns in data. Consequently, researchers have begun exploring the potential of transformers in traditional machine learning tasks, particularly tabular classification. Tabular data represents one of the most fundamental and critical types of information encountered in real-world applications, spanning domains such as recommendation systems (Zhang et al., 2019), finance (Arun et al., 2016), and medicine (Johnson et al., 2016).

Numerous efforts have been made to adapt Transformers for tabular classification tasks (Arik & Pfister, 2021; Hollmann et al., 2023; Huang et al., 2020; Dinh et al., 2022; Gorishniy et al., 2021). For instance, FT-Transformer (Gorishniy et al., 2021) introduces a feature tokenizer to convert each example into a sequence of embeddings, then utilizes a Transformer to process these and make predictions via a special `CLS` token. TabTransformer (Huang et al., 2020) employs the Transformer architecture to learn embeddings for categorical features, concatenating them with continuous features for improved accuracy. LIFT (Dinh et al., 2022) converts tabular datasets into sentences that include feature names and task descriptions, utilizing fine-tuned large language models for predictions. Unfortunately, these aforementioned methods, along with non-Transformer neural network approaches (e.g., Multilayer Perceptron (Rumelhart et al., 1986) and ResNet (He et al., 2016)), suffer from a common inefficiency compared to gradient-boosted trees methods. Their large model sizes result in longer training and inference times.

As a Transformer-based method, TABPFN (Hollmann et al., 2023) stands out for its superior performance and efficiency on small datasets. It leverages a key capability of LLMs: in-context learning (ICL) (Brown et al., 2020), which enables LLMs to learn from a few examples and make predictions for new test instances without needing parameter updates. TABPFN employs a customized

ICL implementation that processes all training and testing samples in a single prompt, completing classification for all test samples in one forward pass. This approach enables rapid predictions within seconds for simple, small tabular datasets, making it highly efficient and effective on such tasks. However, TABPFN faces challenges with complex datasets that typically demand larger sample sizes for effective learning, primarily due to scalability limitations imposed by the quadratic complexity of the attention mechanism. This constraint introduces difficulties in both scalable pre-training and inference processes.

In this paper, we address the scalability limitations of TABPFN and enhance the competitiveness of neural network-based methods for tabular classification. In doing so, we investigate scalable alternatives to traditional attention mechanisms, focusing on state-space models (SSMs), including the recently popular Mamba model (Gu & Dao, 2024), and linear attention (Katharopoulos et al., 2020). Our analysis reveals that **(Finding 1)** the inherent causality of SSMs impedes ICL performance compared to non-causal mechanisms. In contrast, **(Finding 2)** linear attention does not suffer from this limitation, maintaining comparable performance while improving computational efficiency. Based on these findings, we develop our model, TABFLEX, which leverages linear attention. It comprises three sub-models, each optimized for different scenarios, with the most suitable one selected based on dataset characteristics (e.g., sample size). This model supports thousands of features, hundreds of classes, and millions of samples. We conduct comprehensive experiments with TABFLEX across a diverse range of datasets, including small, large, and high-dimensional datasets. **(Finding 3)** TABFLEX demonstrates robust performance with impressive computational efficiency. Notably, on the `poker-hand` dataset, which contains over one million samples, TABFLEX classifies all instances in *less than 5 seconds* while achieving competitive performance. Furthermore, beyond traditional tabular datasets, TABFLEX can also label all samples of MNIST (LeCun et al., 2010) and Fashion-MNIST (Xiao et al., 2017) in less than one second. This highlights TABFLEX as a pioneering approach towards accelerating Transformer-based models for high-dimensional and large-scale datasets, with promising potential for further advancements.

## 2    RELATED WORKS

**Transformer-based Approaches for Tabular Classification.**    Recent years have witnessed numerous attempts to employ Transformers for tabular classification (Arik & Pfister, 2021; Huang et al., 2020; Gorishniy et al., 2021; Dinh et al., 2022; Hollmann et al., 2023). These methods utilize Transformers in diverse ways to tackle tabular data. TabNet (Arik & Pfister, 2021), one of the pioneering efforts, applies unsupervised pre-training on masked tabular datasets to infer missing features, thereby enhancing the model's understanding of datasets and features. It then performs supervised learning on feature selection to obtain the final decision boundary, akin to decision trees. Huang et al. (2020) introduced TabTransformer, which leverages Transformers to better handle categorical features by concatenating their contextual embeddings with numerical features. FT-Transformer (Gorishniy et al., 2021) introduces a feature tokenizer to convert each example into a sequence of embeddings, enabling Transformers to process tabular datasets and make predictions. LIFT (Dinh et al., 2022) utilizes a pre-trained language model with parameter-efficient fine-tuning, incorporating task descriptions and converting each sample into a complete sentence with feature names in the prediction prompt. TABPFN (Hollmann et al., 2023) is trained offline on synthetic datasets derived from prior distributions and performs ICL rather than additional parameter tuning for a given dataset, enabling it to solve small tabular classification tasks within seconds. Prior to our work, TuneTable (Feuer et al., 2024) extended TABPFN to scale to large datasets by performing prefix-tuning for each dataset to achieve better performance. Notably, while most of these methods are computationally intensive due to the need for training large models, TABPFN achieves efficiency through ICL. Our method builds upon TABPFN, extending its scalability to large datasets while maintaining and even improving its efficiency.

**Attention Mechanisms and Scalable Alternatives.**    While attention in Transformers (Vaswani et al., 2017) is central to the strong performance of language models, it encounters scaling challenges for long sequences due to its quadratic computational and memory complexity. To overcome these limitations, several scalable alternatives have been proposed (Gu & Dao, 2024; Dao & Gu, 2024; Katharopoulos et al., 2020; Peng et al., 2023; Orvieto et al., 2023; Sun et al., 2023), all aiming to achieve subquadratic time complexity. Classical RNNs offer one potential solution, providing

efficient linear-time inference. However, they struggle with training efficiency and lack the parallelization capabilities of Transformer architectures. Linear attention (Katharopoulos et al., 2020) addresses both concerns by reformulating self-attention as a linear dot-product of kernel feature maps, reducing the computational complexity from quadratic to linear time. Additionally, causal linear attention can be interpreted as a form of RNN, as the model makes predictions based on a current token and a "hidden state," which summarizes information from the previous tokens. State-space models (SSMs), another popular variant of RNNs, address the drawbacks of classical RNNs by considering linear RNNs and proposing novel algorithms for efficient training (Gu et al., 2021; 2022; Gu & Dao, 2024; Dao & Gu, 2024; Peng et al., 2023; Orvieto et al., 2023; Sun et al., 2023).

Dao et al. (2022) identified that another bottleneck in attention mechanisms' speed stems from the relatively slow access to high-bandwidth memory (HBM) in GPUs. To address this limitation, FlashAttention (Dao et al., 2022; Dao, 2024; Shah et al., 2024) restructures attention computation to optimize the utilization of high-speed on-chip SRAM while minimizing access to slower HBM, thereby enhancing the efficiency of GPU-based attention operations. FlashAttention strategically balances computational efficiency against memory bandwidth efficiency. Although the computational complexity in terms of sequence length remains quadratic, the optimizations introduced by FlashAttention significantly accelerate attention computation in wall-clock time.

We provide extended related works in Sec. A, which offers an in-depth discussion of other baselines, encompassing classical machine learning methods, gradient-boosting decision trees, and non-transformer neural network architectures tailored for tabular classification tasks.

## 3 BACKGROUND

This section elucidates the key concepts underpinning TABPFN and introduces two prominent scalable alternatives to standard attention mechanisms: SSMs and linear attention.

**Implementation of ICL in TabPFN (Hollmann et al., 2023).** To elucidate the efficiency of TABPFN and its ability to classify all samples in a single forward pass, we first describe its ICL implementation. Fig. 1 illustrates how TABPFN processes an entire dataset, classifying all test samples simultaneously. The key innovation lies in treating each sample as a token. The input sequence begins with a concatenation of all training samples, where both features and labels are projected into embeddings using MLPs. Following the training samples, all test samples (features only) are appended, with their features similarly embedded. This concatenated sequence of embeddings is then fed into multiple Transformer layers. Importantly, the outputs corresponding to training sample positions are computed by attending to all other training samples, while the outputs for test sample positions also attend to the training samples — enabling each test prediction to leverage the full training set without being influenced by other test samples. Finally, predictions of the test samples are generated by projecting the Transformer outputs at test positions into probability distributions. This implementation is functionally equivalent to standard ICL

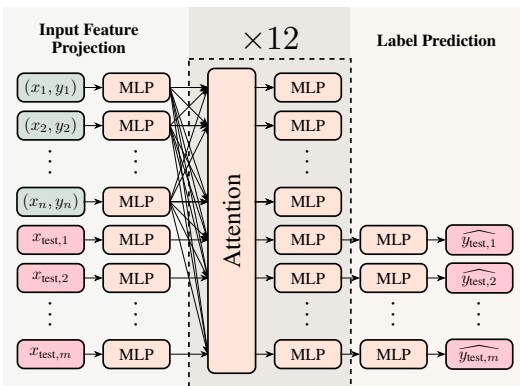

Figure 1: **Illustration of TABPFN's classification approach for an entire dataset via one single forward pass.** In each layer, attention outputs for training sample positions attend to all other training samples, ensuring that predictions are invariant to the order of training samples. Conversely, attention outputs for test sample positions attend only to training samples, ensuring independent predictions for each test instance, unaffected by other test samples. The final classification for each test sample is derived by applying an MLP to the corresponding Transformer output at its respective position.

but significantly more efficient. Standard ICL would require $m$ separate prompts (where $m$ is the number of test samples), each containing all training samples and one test sample, necessitating $m$ prediction passes. A notable feature of TABPFN's architecture is its use of an encoder with non-

causal attention. This allows outputs within training sample positions to interact freely, rendering the order of training samples inconsequential.

**State Space Models (SSMs).**    Recently, SSMs have emerged as highly promising alternatives to the attention mechanism, exhibiting linear computational complexity and demonstrating excellent performance in language modeling tasks. The SSM framework is based on a continuous system that transforms a one-dimensional signal $x(t) \in \mathbb{R}$ into $y(t) \in \mathbb{R}$ through an intermediate $H$-dimensional latent state $\boldsymbol{h}(t) \in \mathbb{R}^H$, as shown in (1). Here, $\boldsymbol{B} \in \mathbb{R}^{H \times 1}$ is the input transition vector and $\boldsymbol{A} \in \mathbb{R}^{H \times H}$ is the state transition matrix. The latent state $\boldsymbol{h}(t)$ is then projected into the output $y(t)$ using the output mapping vector $\boldsymbol{C} \in \mathbb{R}^{1 \times H}$. For deep learning applications, discrete $\overline{\boldsymbol{A}}$ and $\overline{\boldsymbol{B}}$ replace continuous $\boldsymbol{A}$ and $\boldsymbol{B}$ through discretization methods, such as zero-order hold. This yields updated hidden state and output equations as shown in (2). While (2) is structured as linear RNN, it can be reformulated as Convolutional Neural Network (CNN) as (3), enabling efficient and parallelizable training. SSMs address the quadratic time complexity problem with respect to sequence length, as the output for each new token depends solely on the hidden states and the current token, in contrast to standard attention mechanisms that attend to all previous tokens. Consequently, SSMs operate as a causal mechanism.

$$\begin{aligned} \boldsymbol{h}'(t) &= \boldsymbol{A}\boldsymbol{h}(t) + \boldsymbol{B}x(t) \\ y(t) &= \boldsymbol{C}\boldsymbol{h}(t) \end{aligned} \quad (1) \qquad \begin{aligned} \boldsymbol{h}_t &= \overline{\boldsymbol{A}}\boldsymbol{h}_{t-1} + \overline{\boldsymbol{B}}x_t, \\ y_t &= \boldsymbol{C}\boldsymbol{h}_t \end{aligned} \quad (2) \qquad \begin{aligned} \overline{\boldsymbol{K}} &= (\boldsymbol{C}\overline{\boldsymbol{B}}, \boldsymbol{C}\overline{\boldsymbol{A}}\,\overline{\boldsymbol{B}}, \dots, \boldsymbol{C}\overline{\boldsymbol{A}}^{t-1}\overline{\boldsymbol{B}}), \\ (y_1, \dots, y_t) &= (x_1, \dots, x_t) * \overline{\boldsymbol{K}} \end{aligned} \quad (3)$$

**Linear attention.**    Assume a sequence with length $n \in \mathbb{N}^+$ and embedding size $d \in \mathbb{N}^+$. We first focus on non-causal cases. For the $i$-th position, let $\boldsymbol{q}_i \in \mathbb{R}^d$, $\boldsymbol{k}_i \in \mathbb{R}^d$, and $\boldsymbol{v}_i \in \mathbb{R}^d$ denote the query, key, and value vectors, respectively, where $i = 1, \dots, n$. In softmax attention, the similarity between $\boldsymbol{q}_i$ and $\boldsymbol{k}_j$ for any $i \neq j$ is computed as $\exp(\boldsymbol{q}_i^\top \boldsymbol{k}_j)$. The attention output at the $i$-th position, denoted as $\boldsymbol{a}_i \in \mathbb{R}^d$, is obtained by averaging the values across all tokens weighted by their similarities. This process requires $O(n)$ complexity, as it necessitates computing similarities with all $n$ tokens. Linear attention reduces this complexity by replacing the similarity computation from $\exp(\boldsymbol{q}_i^\top \boldsymbol{k}_j)$ with $\phi(\boldsymbol{q}_i)^\top \phi(\boldsymbol{k}_j)$, where $\phi : \mathbb{R}^d \to \mathbb{R}^d$ is a feature conversion function. For linear attention outputs (4) across all positions, we identify two common terms: $\sum_{j=1}^n \phi(\boldsymbol{k}_j) \cdot \boldsymbol{v}_j$ and $\sum_{j=1}^n \phi(\boldsymbol{k}_j)$, which can be computed once. Consequently, for the linear output at position $i$, we only need to compute $\phi(\boldsymbol{q}_i)$ and multiply it with these two statistics, resulting in $O(1)$ complexity, thus significantly reducing computational demands.

$$(\text{Softma}\times) \; \boldsymbol{a}_i = \frac{\sum_{j=1}^n \exp\left(\boldsymbol{q}_i^\top \boldsymbol{k}_j\right) \cdot \boldsymbol{v}_j}{\sum_{j=1}^n \exp\left(\boldsymbol{q}_i^\top \boldsymbol{k}_j\right)} \qquad (\text{Linear}) \; \boldsymbol{a}_i = \frac{\sum_{j=1}^n \phi(\boldsymbol{q}_i)^\top \phi(\boldsymbol{k}_j) \cdot \boldsymbol{v}_j}{\sum_{j=1}^n \phi(\boldsymbol{q}_i)^\top \phi(\boldsymbol{k}_j)} = \frac{\phi(\boldsymbol{q}_i)^\top \sum_{j=1}^n \phi(\boldsymbol{k}_j) \cdot \boldsymbol{v}_j}{\phi(\boldsymbol{q}_i)^\top \sum_{j=1}^n \phi(\boldsymbol{k}_j)}$$

$$(4)$$

For causal cases, for position $i$, we simply replace the sum from $j = 1$ to $n$ with $j = 1$ to $i$, as each token attends only to previous tokens. The statistics then become $\sum_{j=1}^{i-1} \phi(\boldsymbol{k}_j) \cdot \boldsymbol{v}_j$ and $\sum_{j=1}^{i-1} \phi(\boldsymbol{k}_j)$, which can be viewed as hidden states in RNNs. Thus, causal linear attention can be conceptualized as a linear RNN, which is also a variant of SSM.

## 4 ARCHITECTURAL EXPLORATION FOR SCALABLE TABULAR LEARNING

This section examines alternative model architectures to enhance the scalability of the standard attention mechanism used in TABPFN. Among the various options, two primary contenders emerge: (i) State-Space Models (SSMs) and (ii) linear attention. We note that linear attention with causal masking can be viewed as a type of SSM. Our analysis focuses on determining which of these approaches is most effective for tabular classification tasks within the framework of ICL.

### 4.1 CAUSAL MODEL VS. NON-CAUSAL MODEL

Ideally, the order of training samples (i.e., in-context demonstrations) provided in the prompt should not influence the final prediction. However, SSMs are inherently causal, computing outputs based on new inputs and hidden states derived from previous inputs. This characteristic suggests a potential drawback for SSMs in this context. To validate our hypothesis regarding the suboptimal

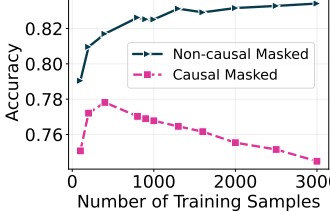 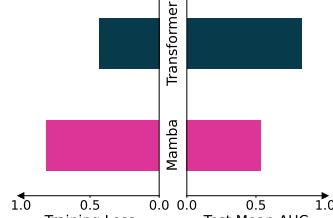 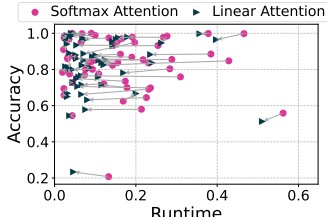

(a) Effect of causal masking on performance. Non-causal model shows better sample utilization and accuracy as the number of samples grows. In contrast, causal model's performance plateaus early and declines as more samples are added.

(b) ICL performance comparison between Mamba and Transformer models. Results show Transformer-based models achieve lower training loss and higher AUC across 150 test datasets.

(c) Accuracy and runtime comparison of softmax and linear attention. Replacing softmax with linear attention preserves comparable accuracy while significantly reducing runtime.

Figure 2: Impact of model architecture on tabular classification performance.

performance of causal models in ICL, we conduct two experiments: (i) we compare the performance of TABPFN with a modified version of the same model that uses causal attention, and (ii) we evaluate TABPFN against both its original version and a model incorporating Mamba (specifically Mamba-II), a leading SSM-based architecture.

**Causal Attention vs. Non-Causal Attention.** In our first experiment, we compare the ICL capabilities of non-causal and causal attention mechanisms using the same experimental setup as TABPFN. We replicate TABPFN's methodology for generating synthetic datasets from priors, training a modified version of TABPFN that employs causal attention instead. For the inference stage, we generate 20 synthetic datasets. Each dataset maintains a consistent 1000 test samples while we vary the number of training samples. We then calculate the classification accuracy for each dataset and average the results across all 20 simulations. The results are visualized in Fig. 2a.

Our observations reveal that non-causal attention generally outperforms causal attention. As we increase the number of training samples, the accuracy of the non-causal model continues to improve. In contrast, the causal attention model shows accuracy improvements only within a very small range of training samples, after which performance begins to decline with additional samples. These findings indicate that TABPFN with non-causal attention functions as an effective ICL model, adeptly leveraging context from a large number of samples. Conversely, the same model equipped with causal attention fails to capitalize on the additional data, highlighting the superiority of the non-causal approach in this tabular learning scenario.

**Mamba vs. Transformer.** In this experiment, we further investigate whether Mamba, the most popular SSM-based model, is suitable for ICL. We replicate TABPFN's training methodology precisely, substituting the transformer layer with a Mamba layer. To evaluate performance, we test the modified model on the same 150 validation datasets used in the original TABPFN study (refer to Section F.3 of their paper for details). Fig. 2b visualizes the training loss and test mean AUC for both methods. We observe that the model with Mamba exhibits significantly higher training loss compared to the original TABPFN, along with substantially lower test mean AUC. This experiment with a popular SSM model further demonstrates that SSMs underperform non-causal models in our specified tasks.

## 4.2 SOFTMAX ATTENTION VS. LINEAR ATTENTION

To address the quadratic complexity of standard attention mechanisms, linear attention has emerged as a popular alternative (Katharopoulos et al., 2020). To investigate its impact on ICL in tabular classification, we replaced TABPFN's attention mechanism with linear attention and trained a model following the same strategy as TABPFN. We then evaluated both TABPFN and this linear attention model on 57 real datasets (used in Table 2 of McElfresh et al. (2023), where TABPFN achieved top performance among 19 methods for tabular classification). Fig. 2c visualizes the test accuracy and runtime. Our results demonstrate that linear attention does not decrease performance and significantly improves speed, making it a suitable method for scaling TABPFN to larger datasets. To better understand the strong performance of linear attention in in-context learning, we provide a detailed discussion in Sec. A. Furthermore, in Sec. B, we investigate the use of sample selection to further accelerate tabular classification. Finally, in Sec. C, we demonstrate that linear attention significantly outperforms sliding window attention (Beltagy et al., 2020) in our setting.

---

**Algorithm 1:** Conditional Model Selection

---
**Input:** A dataset $\mathcal{D}$ with number of instances $n$ and number of features $d$

1 // Large dataset with few features;
2 **if** $n \geq 3K$ *and* $d \leq 100$ **then**
3      **return** TABFLEX-L100($\mathcal{D}$);
4 // High-dimensional datasets;
5 **else if** $d > 100$ *or* ($d/n \geq 0.2$ *and* $n \geq 3K$) **then**
6      **if** $d \leq 1000$ **then**
7          **return** TABFLEX-H1K($\mathcal{D}$);
8      **else**
9          Apply random projection to $\mathcal{D}$ to reduce the number of features to 1000, yielding $\mathcal{D}'$;
10          **return** TABFLEX-H1K($\mathcal{D}'$);
11 // Small datasets;
12 **else**
13      **return** TABFLEX-S100($\mathcal{D}$);

---

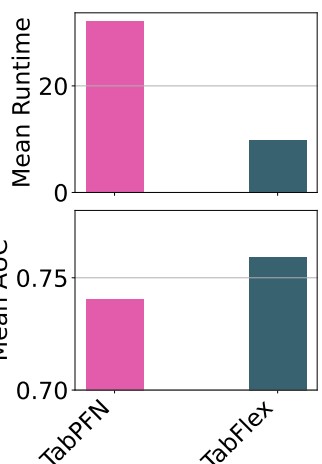

Figure 3: Runtime and AUC comparison of TABPFN and TABFLEX on validation datasets.

# 5 TABFLEX: SCALING TABPFN FOR LARGE DATASETS

Based on the empirical findings presented in Sec. 4, we identify non-causal linear attention as the optimal candidate to replace standard softmax attention in TABPFN. This section proceeds in two parts: first, we conduct a thorough analysis of the linear attention mechanism to ensure its efficient implementation.; subsequently, we leverage this efficient implementation to train our proposed model, TABFLEX. Our approach aims to enhance the scalability and performance of tabular learning while maintaining computational efficiency.

**Computation Analysis.** Dao et al. (2022) demonstrates that significant wallclock speedup for softmax attention can be achieved by optimizing the number of memory reads/writes between GPU high bandwidth memory (HBM) and GPU on-chip SRAM. Based on this criterion, Yang et al. (2024) proposed FlashLinearAttention for speeding up *causal* linear attention. This raises a natural question: can we further improve the speed of non-causal linear attention (we omit non-causal when it does not cause further confusion) by reducing the number of memory reads/writes? Our results in Theorem 1 analyze the #HBM access and HBM memory usage of FlashLinearAttention and linear attention, concluding that further optimization is not necessary. In Sec. D, we first propose an HBM-efficient linear attention, and then show that the PyTorch implementation only incurs a marginal increase in terms of #HBM access and HBM memory usage, with FLOPS remaining unchanged. We provide more details, including the analysis of different attention mechanisms and actual memory usage and runtime visualization of these mechanisms in Sec. D. The resulting theorem below demonstrates that the straightforward PyTorch implementation of linear attention already achieves linear HBM access, matching the performance of FlashLinearAttention after optimization. Consequently, we adopt the straightforward implementation of linear attention in our model.

**Theorem 1** (High Bandwidth Memory Efficiency of Linear Attention). *Let $\boldsymbol{Q}, \boldsymbol{K}, \boldsymbol{V} \in \mathbb{R}^{N \times D}$ represent the query, key, and value matrices for a single attention head, where $N$ is the sequence length and $D$ is the embedding size. Both causal FlashLinearAttention (Alg. 2) and non-causal linear attention (Listing 1) require $O(ND)$ HBM accesses, $O(ND)$ HBM memory, and $O(ND^2)$ FLOPS to compute the attention output.*

**TABFLEX.** While TABPFN excels on small, simple datasets with fewer than 100 features and 10 classes, it struggles with more complex tasks, such as high-dimensional datasets or those with numerous classes. Our objective is to extend the use cases by training a model that maintains comparable speed to TABPFN while offering reasonable performance across a broader spectrum of datasets. Since models trained with numerous features and long contexts often suffer from poor performance in small regions due to optimization challenges, we develop three specialized models:

- **TABFLEX-S100**: Trained on prompts with 1152 length (same as TABPFN), 100 features, and 10 classes. Optimized for low-dimensional datasets. 'S' denotes standard configuration, '100' indicates feature capacity.

| Algorithm | Class | Mean AUC | | Std. AUC | | Time / 1000 inst. | |
|---|---|---|---|---|---|---|---|
| | | median | mean | mean | median | median | mean |
| TabPFN (Hollmann et al., 2023) | TF | 0.97 | 0.84 | 0.15 | 0.08 | 0.56 | 0.74 |
| CatBoost (Prokhorenkova et al., 2018) | GBDT | 0.97 | 0.92 | 0.15 | 0.07 | 1.95 | 20.51 |
| **TABFLEX (Ours)** | TF | 0.96 | 0.90 | 0.15 | 0.08 | 0.22 | 0.37 |
| XGBoost (Chen & Guestrin, 2016) | GBDT | 0.96 | 0.91 | 0.16 | 0.09 | 0.38 | 0.85 |
| RandomForest (Liaw et al., 2002) | Classical | 0.95 | 0.90 | 0.16 | 0.09 | 0.32 | 0.47 |
| SAINT (Somepalli et al., 2021) | TF | 0.94 | 0.86 | 0.16 | 0.11 | 146.15 | 170.56 |
| HyperFast (Bonet et al., 2024) | Non-TF NN | 0.94 | 0.87 | 0.15 | 0.09 | 53.45 | 89.75 |
| LightGBM (Ke et al., 2017) | GBDT | 0.93 | 0.85 | 0.18 | 0.09 | 0.29 | 0.90 |
| ResNet (He et al., 2016) | Non-TF NN | 0.93 | 0.85 | 0.16 | 0.10 | 8.83 | 15.99 |
| DANet (Chen et al., 2022) | Non-TF NN | 0.92 | 0.85 | 0.16 | 0.08 | 57.18 | 64.29 |
| NODE (Popov et al., 2019) | Non-TF NN | 0.91 | 0.83 | 0.16 | 0.11 | 131.73 | 160.76 |
| FTTransformer (Gorishniy et al., 2021) | TF | 0.89 | 0.81 | 0.17 | 0.11 | 18.04 | 27.91 |
| SVM (Cortes, 1995) | Classical | 0.89 | 0.78 | 0.19 | 0.09 | 2.06 | 61.18 |
| MLP-rtdl (Gorishniy et al., 2021) | Non-TF NN | 0.88 | 0.75 | 0.18 | 0.11 | 7.09 | 15.21 |
| DeepFM (Guo et al., 2017) | Non-TF NN | 0.87 | 0.77 | 0.19 | 0.12 | 4.89 | 6.05 |
| TabNet (Arik & Pfister, 2021) | TF | 0.85 | 0.68 | 0.26 | 0.14 | 29.34 | 35.12 |
| STG (Yamada et al., 2020) | Non-TF NN | 0.82 | 0.71 | 0.20 | 0.14 | 15.98 | 18.58 |
| TuneTables (Feuer et al., 2024) | TF | 0.81 | 0.70 | 0.25 | 0.16 | 32.96 | 73.40 |
| LinearModel (Cox, 1958) | Classical | 0.78 | 0.67 | 0.19 | 0.14 | 0.03 | 0.04 |
| MLP (Rumelhart et al., 1986) | Non-TF NN | 0.76 | 0.68 | 0.20 | 0.13 | 11.23 | 18.31 |
| DecisionTree (Quinlan, 1986) | Classical | 0.74 | 0.63 | 0.24 | 0.18 | 0.01 | 0.03 |
| TabTransformer (Huang et al., 2020) | TF | 0.72 | 0.61 | 0.17 | 0.13 | 13.45 | 22.05 |
| KNN (Cover & Hart, 1967) | Classical | 0.70 | 0.61 | 0.21 | 0.14 | 0.03 | 0.05 |
| VIME (Yoon et al., 2020) | Non-TF NN | 0.60 | 0.54 | 0.25 | 0.15 | 15.60 | 17.98 |
| NAM (Agarwal et al., 2021) | Non-TF NN | 0.39 | 0.44 | 0.27 | 0.19 | 97.99 | 233.77 |

Table 1: **Performance comparison of algorithms across 98 simple datasets (as used in Table 1 of** McElfresh et al. **(2023)**). The reported AUC values are normalized. The "Time/1000 inst." column represents the combined training and test time for all datasets, divided by the total number of samples. Notably, TABFLEX achieves top 3 performance, with faster runtimes compared to baselines of similar performance, and a 2× speedup relative to TABPFN.

- **TABFLEX-L100**: Utilizes prompts of 50K length, 100 features, and 10 classes. Designed for large low-dimensional datasets. 'L' signifies larger sample size, '100' represents feature count.

- **TABFLEX-H1K**: Employs prompts of 50K length, 1K features, and 100 classes. Suited for large high-dimensional datasets. 'H' indicates high-dimensional capabilities, '1K' denotes 1K features.

We use a conditional model selection strategy, as shown in the Alg. 1, to choose the appropriate model based on the target dataset's size and dimensionality, ensuring optimal performance across diverse data characteristics. Our code is publicably accessible at `https://anonymous.4open.science/r/tabflex`. Additional training details, including training loss, hyperparameters, and other relevant information, are provided in Sec. E.1.

In Fig. 3, we visualize the mean runtime and mean AUC comparison of TABPFN and TABFLEX on the validation datasets, comprising 40 datasets with varying sample sizes (up to 100K), dimensions (up to 3K), and number of classes (up to 100). Detailed information about these datasets is provided in Sec. E.2. Our analysis reveals that TABFLEX not only exhibits superior performance but also demonstrates faster execution times compared to TABPFN.

# 6 EXPERIMENTS

In this section, we evaluate TABFLEX's performance and speed across 115 OpenML tabular datasets (Vanschoren et al., 2013). Our results show that TABFLEX achieves comparable performance to TABPFN on small datasets while offering significant speedup, and substantially outperforms it on high-dimensional and large datasets. TABFLEX exhibits competitive performance among 23 common baselines while maintaining high efficiency, notably processing the largest dataset with over one million samples in just 4.88 seconds.

## 6.1 EXPERIMENTAL SETUP

Unless otherwise stated, we follow the identical experiment setup of McElfresh et al. (2023) for benchmarking all baselines.

| Algorithm | Class | Mean AUC | | Std. AUC | | Time / 1000 inst. | |
|---|---|---|---|---|---|---|---|
| | | median | mean | mean | median | median | mean |
| TabPFN (Hollmann et al., 2023) | TF | 0.97 | 0.90 | 0.21 | 0.15 | 0.82 | 1.04 |
| **TABFLEX (Ours)** | TF | 0.96 | 0.89 | 0.22 | 0.16 | 0.29 | 0.48 |
| CatBoost (Prokhorenkova et al., 2018) | GBDT | 0.95 | 0.89 | 0.23 | 0.16 | 2.59 | 19.51 |
| ResNet (He et al., 2016) | Non-TF NN | 0.93 | 0.84 | 0.24 | 0.16 | 13.90 | 23.40 |
| SAINT (Somepalli et al., 2021) | TF | 0.93 | 0.84 | 0.24 | 0.20 | 173.63 | 195.16 |
| RandomForest (Liaw et al., 2002) | Classical | 0.92 | 0.86 | 0.24 | 0.17 | 0.45 | 0.61 |
| XGBoost (Chen & Guestrin, 2016) | GBDT | 0.91 | 0.86 | 0.24 | 0.18 | 0.49 | 0.95 |
| HyperFast (Bonet et al., 2024) | Non-TF NN | 0.91 | 0.83 | 0.22 | 0.17 | 64.38 | 136.74 |
| DANet (Chen et al., 2022) | Non-TF NN | 0.89 | 0.80 | 0.25 | 0.19 | 67.70 | 78.21 |
| SVM (Cortes, 1995) | Classical | 0.87 | 0.75 | 0.28 | 0.22 | 0.71 | 87.84 |
| NODE (Popov et al., 2019) | Non-TF NN | 0.86 | 0.80 | 0.24 | 0.18 | 157.18 | 194.07 |
| DeepFM (Guo et al., 2017) | Non-TF NN | 0.86 | 0.79 | 0.28 | 0.27 | 5.48 | 5.95 |
| FTTransformer (Gorishniy et al., 2021) | TF | 0.84 | 0.78 | 0.25 | 0.21 | 25.40 | 33.34 |
| LightGBM (Ke et al., 2017) | GBDT | 0.83 | 0.76 | 0.28 | 0.21 | 0.25 | 0.67 |
| MLP-rtdl (Gorishniy et al., 2021) | Non-TF NN | 0.83 | 0.74 | 0.26 | 0.20 | 12.65 | 22.97 |
| LinearModel (Cox, 1958) | Classical | 0.81 | 0.71 | 0.27 | 0.21 | 0.05 | 0.06 |
| TuneTables (Feuer et al., 2024) | TF | 0.80 | 0.72 | 0.32 | 0.24 | 53.48 | 113.49 |
| STG (Yamada et al., 2020) | Non-TF NN | 0.79 | 0.67 | 0.29 | 0.23 | 18.46 | 21.26 |
| TabTransformer (Huang et al., 2020) | TF | 0.79 | 0.64 | 0.24 | 0.16 | 19.04 | 32.84 |
| MLP (Rumelhart et al., 1986) | Non-TF NN | 0.72 | 0.65 | 0.29 | 0.25 | 17.83 | 27.67 |
| DecisionTree (Quinlan, 1986) | Classical | 0.63 | 0.55 | 0.35 | 0.31 | 0.01 | 0.02 |
| KNN (Cover & Hart, 1967) | Classical | 0.62 | 0.56 | 0.30 | 0.25 | 0.03 | 0.03 |
| TabNet (Arik & Pfister, 2021) | TF | 0.56 | 0.50 | 0.42 | 0.40 | 34.66 | 42.09 |
| VIME (Yoon et al., 2020) | Non-TF NN | 0.49 | 0.48 | 0.37 | 0.27 | 18.43 | 20.11 |
| NAM (Agarwal et al., 2021) | Non-TF NN | 0.33 | 0.38 | 0.38 | 0.31 | 147.30 | 341.58 |

Table 2: **Performance of algorithms across 57 datasets of size less than or equal to 1250 (used in Table 2 of McElfresh et al. (2023)).** The reported AUC values are normalized. The "Time/1000 inst." column represents the combined training and test time for all datasets, divided by the total number of samples. Notably, TABFLEX achieves top 2 performance, with significant faster runtimes compared to baselines of similar performance, and a 2× speedup relative to TABPFN.

**Datasets.** For simple datasets, we use two sets of datasets, the first one include 98 datasets reported in Table 1 of McElfresh et al. (2023), while the second one include 57 datasets reported in Table 2 of McElfresh et al. (2023). Lastly, we evaluate the methods on the TabZilla hard benchmark (McElfresh et al., 2023), which comprises 36 challenging datasets, including 11 high-dimensional (with $100 \leq$ features $\leq 2000$) and large (containing $\geq 50K$ instances) datasets. Detailed information about the datasets, including their names and characteristics, is provided in Sec. F.1. Furthermore, we consider additional datasets, with details and results presented in Sec. F.2.

**Baselines.** We evaluate our approach against a comprehensive set of baselines, as considered by McElfresh et al. (2023). These include: (i) classical methods: Random Forest (Liaw et al., 2002), SVM (Cortes, 1995), LinearModel (Cox, 1958), KNN (Cover & Hart, 1967) and Decision Tree (Quinlan, 1986); (ii) Gradient Boosted Decision Trees (GBDT) methods: XGBoost (Chen & Guestrin, 2016), CatBoost (Prokhorenkova et al., 2018), and LightGBM (Ke et al., 2017); (iii) Non-Transformer Neural Network (Non-TF NN) methods: SAINT (Somepalli et al., 2021), ResNet (He et al., 2016), DANet (Chen et al., 2022), NODE (Popov et al., 2019), MLP (Rumelhart et al., 1986), MLP-rtdl (Gorishniy et al., 2021), DeepFM (Guo et al., 2017), STG (Yamada et al., 2020), VIME (Yoon et al., 2020), and NAM (Agarwal et al., 2021); (iv) Transformer (TF) methods: TABPFN (Hollmann et al., 2023), FTTransformer (Gorishniy et al., 2021), TabNet (Arik & Pfister, 2021), and TabTransformer (Huang et al., 2020). The results for these methods, except TABPFN, are taken directly from McElfresh et al. (2023), who conducted their experiments using a V100 GPU, while our experiments are run on an A100 GPU, which may introduce slight variations in performance. Additionally, we incorporate two recent methods designed for scaling tabular classification: TuneTables (Feuer et al., 2024), a TF method, and HyperFast (Bonet et al., 2024), a Non-TF NN method.

Note that not all baselines successfully ran on all datasets. Many methods face constraints and encounter issues, particularly with the TabZilla hard benchmark, often due to poor scalability. We explicitly indicate which methods failed to run smoothly across all datasets. Originally, TABPFN was limited to datasets with no more than 100 features and 10 classes. To facilitate a fair comparison between TABFLEX and TABPFN, we implemented workarounds to prevent TABPFN from encountering errors. For datasets exceeding 100 features, we performed random feature selection. For those with more than 10 classes, we evaluated the accuracy of the nine most prevalent classes and marked all other classes as other, and incorrect. For TuneTables, we directly import

`TuneTablesClassifier` from their Python package `tunetables`. Note that our results differ from those reported in their paper, as their study involved more extensive hyperparameter search, which significantly increased runtime. We also compare our methods with TuneTables using the dataset split specified in their paper's setting, with results deferred to Sec. F.3. Similarly, for HyperFast, we utilize `HyperFastClassifier` directly from their Python package `hyperfast` default parameters. Notably, HyperFast is meta-trained on many datasets we use for evaluation.

## 6.2 EVALUATION ON SIMPLE DATASETS

We evaluate TABFLEX's tabular classification performance on two sets of datasets: 98 simple datasets from Table 1 and 57 small datasets from Table 2 of McElfresh et al. (2023). The results are reported in Table 1 and Table 2, respectively. For each dataset, we consider ten different train/test splits, computing the mean and standard deviation of AUC, as well as the total runtime per 1000 instances. We then calculate the median and mean of these values across the entire set of datasets: 98 simple datasets for Table 1 and 57 small datasets for Table 2. Algorithms are ranked based on AUC and time. Our results demonstrate that TABFLEX achieves nearly identical performance to TABPFN on small, simple datasets while offering more than a 2x speedup. Compared to faster methods, such as Decision Tree and Linear Model in Table 1, and Decision Tree, Linear Model, LightGBM, and KNN in Table 2, their performance is significantly inferior to TABFLEX.

## 6.3 EVALUATION ON HARD DATASETS

In this experiment, we compare TABFLEX to baselines on the TabZilla hard benchmark (McElfresh et al., 2023), which includes 36 datasets. However, due to the challenging nature of the datasets in the TabZilla hard benchmark, many baselines fail to execute successfully. In Fig. 4, we visualize the Median AUC and the runtime per 1000 instances across the 36 datasets, with methods that successfully executed on all datasets marked as stars, and methods that failed to execute on some datasets marked as circles. This figure focuses on efficient methods, excluding those slower than 0.5 seconds per 1000 instances. We observe that only TABFLEX, TABPFN, and XG-Boost successfully run on all datasets. Notably, TABFLEX is faster and achieves better performance than TABPFN, and is faster than XGBoost while sacrificing only a small margin of performance.

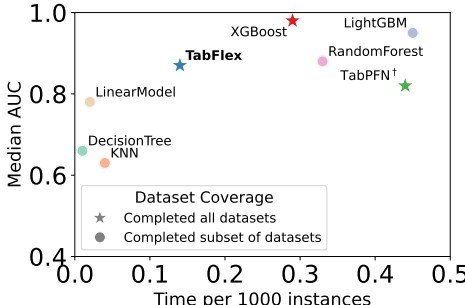

Figure 4: **Visualization of tabular classification methods with processing times under 0.5 seconds per 1000 instances on the TabZilla hard benchmark (McElfresh et al., 2023).** For methods that only completed experiments on a subset of datasets, we report the median AUC across these completed datasets. Compared to two other methods (XGBoost and TABPFN) that successfully ran on all datasets, TABFLEX achieves a 2× speedup while maintaining relatively good performance.

Next, we focus on 11 high-dimensional and large datasets within the TabZilla hard benchmark. Since most baselines do not obtain complete results for all datasets, instead of comparing TABFLEX to a specific baseline, we report the 5th-best AUC and 5th-best runtime, using these values to summarize the general performance distribution of the baselines. The results are presented in Table 3. We observe that, for these datasets, TABFLEX substantially outperforms TABPFN. While TABPFN follows McElfresh et al. (2023)'s strategy of using only 3000 training samples, TABFLEX utilizes all available training data, achieving superior performance with comparable or slightly higher processing times. TABFLEX exhibits competitive performance among baselines while maintaining high efficiency. Notably, on large datasets with more than 50K instances, TABFLEX is significantly faster than the baselines. For instance, on the largest dataset, *poker-hand*, containing over one million samples, TABFLEX significantly outperforms other baselines, classifying all samples in just 4.88 seconds, while the fifth fastest method requires more than 500 seconds.

## 6.4 EXTENDING TABFLEX FOR IMAGE CLASSIFICATION

We explore the application of TABFLEX to image classification tasks, comparing it against MLP and ResNet architecture. Our evaluation uses straightforward configurations without extensive

| Dataset | #Classes | #Features | #Instances | AUC | | | Time (seconds) | | |
|---|---|---|---|---|---|---|---|---|---|
| | | | | 5th Best | TABPFN | TABFLEX | 5th Best | TABPFN | TABFLEX |
| SpeedDating | 2 | 120 | 8378 | 0.86 | 0.55 | 0.85 | 1.58 | 1.58 | 1.89 |
| higgs | 2 | 28 | 98050 | 0.79 | 0.72 | 0.76 | 3.46 | 2.82 | 4.92 |
| cnae-9 | 9 | 856 | 1080 | 1.00 | 0.48 | 0.96 | 0.51 | 0.51 | 3.80 |
| albert | 2 | 78 | 425240 | 0.71 | 0.69 | 0.70 | 33.98 | 9.39 | 13.46 |
| audiology | 24 | 69 | 226 | 0.92 | 0.82 | 0.81 | 0.13 | 0.23 | 0.26 |
| jasmine | 2 | 144 | 2984 | 0.86 | 0.70 | 0.86 | 0.68 | 1.27 | 0.99 |
| nomao | 2 | 118 | 34465 | 0.99 | 0.76 | 0.99 | 4.03 | 1.82 | 5.34 |
| Bioresponse | 2 | 1776 | 3751 | 0.85 | 0.50 | 0.75 | 2.49 | 1.29 | 12.38 |
| MiniBooNE | 2 | 50 | 130064 | 0.98 | 0.98 | 0.97 | 10.80 | 3.19 | 7.22 |
| airlines | 2 | 7 | 539383 | 0.70 | 0.63 | 0.64 | 6.53 | 9.73 | 4.20 |
| poker-hand | 10 | 10 | 1025009 | 0.54 | 0.72 | 0.84 | 504.52 | 15.36 | 4.88 |

Table 3: **Performance comparison of TABFLEX, TABPFN, and other baselines on large, high-dimensional datasets from the TabZilla hard benchmark (McElfresh et al., 2023).** Baseline results are summarized by the 5th highest AUC and 5th lowest runtime for each dataset. TABFLEX significantly outperforms TABPFN on these datasets, achieving comparable performance to other baselines while maintaining exceptional speed.

hyperparameter optimization to maintain reasonable computational costs. The MLP implementations include both two-layer and three-layer variants, each configured with 10 hidden neurons and trained for 70 epochs at a fixed learning rate of 0.001. The ResNet architecture employs 2 residual blocks with main and hidden dimension sizes of 128 and 256, respectively. The experimental results demonstrate that TABFLEX achieves remarkable efficiency gains, operating $30\times$ faster than the MLP and $400\times$ faster than the ResNet while maintaining competitive performance. This represents a significant advancement in image classification efficiency, particularly noteworthy given that previous approaches like TABPFN were constrained to small, low-dimensional datasets. Although our validation on MNIST represents a preliminary step, it establishes a promising foundation for extending these techniques to more complex image classification tasks.

| Dataset | Two-Layer MLP | | Three-Layer MLP | | ResNet | | TABFLEX (Ours) | |
|---|---|---|---|---|---|---|---|---|
| | AUC | Time (s) | AUC | Time (s) | AUC | Time (s) | AUC | Time (s) |
| MNIST | 0.924 | 23.547 (30.5×) | 0.959 | 23.060 (29.9×) | - | - | 0.948 | 0.771 |
| Fashion-MNIST | 0.793 | 23.340 (28.8×) | 0.853 | 23.604 (29.1×) | .990 | 398.45 (491.1×) | 0.979 | 0.810 |

Table 4: Performance comparison of TABFLEX against baseline models on image datasets.

## 7 CONCLUSION & DISCUSSION

**Conclusion.** To extend TABPFN for ICL on larger and more challenging tabular classification tasks, in this paper, we conduct a comprehensive exploration of scalable alternatives to attention, ultimately selecting non-causal linear attention. Through computational analysis for algorithmic optimization of the implementation of linear attention, we develop our model, TABFLEX. We demonstrate that TABFLEX achieves comparable performance to TABPFN on small datasets with more than $2\times$ speedup, while outperforming most other baselines with significantly reduced computational time. Moreover, TABFLEX significantly outperforms TABPFN on larger and more complex datasets, becoming much faster than most other baselines on datasets larger than 100K samples, while maintaining performance on par with state-of-the-art methods. We posit that TABFLEX further elevates the performance ceiling of neural network-based models on tabular classification tasks.

**Limitations & Future Works.** While our work achieves fast inference and relatively well performance on datasets with approximately two thousand features, extending it to scale to more features remains an intriguing research direction. Notably, image classification tasks typically involve a large number of features. Adapting our work for image classification could lead to broader applications, given its extremely fast inference and ability to simultaneously output labels for all test samples, making this a promising avenue for future research. For image classification, one potential approach could involve using a visual encoder to preprocess the images before feeding them into our model — a strategy that may prove effective. Beyond image datasets, extending our work to other modalities such as audio classification is also of interest. This expansion might necessitate developing novel methods for generating synthetic datasets for model pretraining, as well as conducting comprehensive analyses on the impact of various hyperparameters such as the number of layers and embedding size. Such investigations would optimize the model architecture to effectively handle an increased number of features, potentially broadening the applicability of our approach across diverse domains.

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

# Appendix

## A  EXTENDED RELATED WORKS

**Classical Machine Learning Approaches for Tabular Classification.**  Classical machine learning algorithms have long been the foundation of tabular data classification. These methods include k-Nearest Neighbors (KNN) (Cover & Hart, 1967), Logistic Regression (Cox, 1958), Decision Trees (Quinlan, 1986), and Support Vector Machines (SVM) (Cortes, 1995). These classical models, while effective, often struggle to handle complex, high-dimensional tabular datasets, motivating the development of more sophisticated approaches.

**Gradient-Boosting Decision Trees for Tabular Classification**  Gradient-boosting decision trees (GBDTs) (Friedman, 2001) have emerged as a cornerstone in tabular classification, owing to their exceptional ability to capture intricate patterns in structured data. By iteratively combining predictions from weak learners, GBDTs refine their outputs to minimize errors, resulting in high predictive accuracy. XGBoost (Chen & Guestrin, 2016) introduced weighted quantile sketching, advanced regularization techniques, and sparsity-awareness, achieving state-of-the-art performance. Light-GBM (Ke et al., 2017), a computationally efficient GBDT implementation, employs Gradient-based One-Side Sampling and a leaf-wise tree growth strategy. CatBoost (Prokhorenkova et al., 2018) leverages symmetric trees and introduces ordered boosting, with a particular emphasis on effectively handling categorical features. These advancements have rendered GBDTs not only powerful but also versatile tools in the domain of tabular data, dominating tabular classification in terms of both speed and performance until the advent of TABPFN.

**Transformer-based Approaches for Tabular Classification.**  Recent years have witnessed numerous attempts to employ Transformers for tabular classification (Arik & Pfister, 2021; Huang et al., 2020; Gorishniy et al., 2021; Dinh et al., 2022; Hollmann et al., 2023). These methods utilize Transformers in diverse ways to tackle tabular data. TabNet (Arik & Pfister, 2021), one of the pioneering efforts, applies unsupervised pre-training on masked tabular datasets to infer missing features, thereby enhancing the model's understanding of datasets and features. It then performs supervised learning on feature selection to obtain the final decision boundary, akin to decision trees. Huang et al. (2020) introduced TabTransformer, which leverages Transformers to better handle

categorical features by concatenating their contextual embeddings with numerical features. While TabTransformer processes categorical and continuous features separately, SAINT (Somepalli et al., 2021) projects both feature types into a shared embedding space before passing them through transformer blocks, thereby enhancing overall performance. FT-Transformer (Gorishniy et al., 2021) introduces a feature tokenizer to convert each example into a sequence of embeddings, enabling Transformers to process tabular datasets and make predictions. LIFT (Dinh et al., 2022) utilizes a pre-trained language model with parameter-efficient fine-tuning, incorporating task descriptions and converting each sample into a complete sentence with feature names in the prediction prompt. TABPFN (Hollmann et al., 2023) is trained offline on synthetic datasets derived from prior distributions and performs ICL rather than additional parameter tuning for a given dataset, enabling it to solve small tabular classification tasks within seconds. Prior to our work, TuneTable (Feuer et al., 2024) extended TABPFN to scale to large datasets by performing prefix-tuning for each dataset to achieve better performance. Notably, while most of these methods are computationally intensive due to the need for training large models, TABPFN achieves efficiency through ICL. Our method builds upon TABPFN, extending its scalability to large datasets while maintaining and even improving its efficiency.

**Attention Mechanisms and Scalable Alternatives.** While attention in Transformers (Vaswani et al., 2017) is central to the strong performance of language models, it encounters scaling challenges for long sequences due to its quadratic computational and memory complexity. To overcome these limitations, several scalable alternatives have been proposed (Gu & Dao, 2024; Dao & Gu, 2024; Katharopoulos et al., 2020; Peng et al., 2023; Orvieto et al., 2023; Sun et al., 2023), all aiming to achieve subquadratic time complexity. In contrast, classical RNNs provide the advantage of efficient linear-time inference but suffer from limitations in training efficiency, lacking the parallelization capabilities of Transformer architectures. Linear attention (Katharopoulos et al., 2020) addresses both concerns by reformulating self-attention as a linear dot-product of kernel feature maps, reducing the computational complexity from quadratic to linear time. Additionally, causal linear attention can be interpreted as a form of RNN, as the model makes predictions based on a current token and a "hidden state," which summarizes information from the previous tokens. State-space models (SSMs), another popular variant of RNNs, address the drawbacks of classical RNNs by considering linear RNNs and proposing novel algorithms for efficient training (Gu et al., 2021; 2022; Gu & Dao, 2024; Dao & Gu, 2024; Peng et al., 2023; Orvieto et al., 2023; Sun et al., 2023).

Dao et al. (2022) identified that another bottleneck in attention mechanisms' speed stems from the relatively slow access to high-bandwidth memory (HBM) in GPUs. To address this limitation, FlashAttention (Dao et al., 2022; Dao, 2024; Shah et al., 2024) restructures attention computation to optimize the utilization of high-speed on-chip SRAM while minimizing access to slower HBM, thereby enhancing the efficiency of GPU-based attention operations. FlashAttention strategically balances computational efficiency against memory bandwidth efficiency. Although the computational complexity in terms of sequence length remains quadratic, the optimizations introduced by FlashAttention significantly accelerate attention computation in wall-clock time.

**Non-Transformer Neural Network-based Approaches for Tabular Classification.** Non-Transformer neural networks, such as Multi-Layer Perceptrons (MLP) (Rumelhart et al., 1986), were explored for tabular classification long before Transformer-based methods, but their performance was limited. In recent years, several novel neural network techniques have been developed for this task, including ResNet (He et al., 2016), DANet (Chen et al., 2022), NODE (Popov et al., 2019), DeepFM (Guo et al., 2017), STG (Yamada et al., 2020), VIME (Yoon et al., 2020), and NAM (Agarwal et al., 2021). DeepFM (Guo et al., 2017) employs a factorization machine-based neural network to learn from categorical data. Drawing inspiration from CatBoost, Popov et al. (2019) present a novel neural network architecture designed specifically for tabular data, named Neural Oblivious Decision Ensembles (NODE). While self- and semi-supervised learning have demonstrated effectiveness in the domains of computer vision and natural language processing, Yoon et al. (2020) proposed Value Imputation and Mask Estimation (VIME), which represents the first attempt to address tabular tasks using a self- and semi-supervised learning framework. Agarwal et al. (2021) proposed the Neural Additive Model (NAM), an interpretable neural network that maintains strong performance on tabular data. Yamada et al. (2020) proposed a feature selection method using stochastic gates (STG), which is a neural network-based and effective approach for tabular data. Chen et al.

(2022) designed an abstract layer, a specialized neural component for tabular data, and proposed Deep Abstract Networks (DANets) by stacking these layers.

Some approaches even replace Transformers with SSMs for tabular learning (Ahamed & Cheng, 2024; Thielmann et al., 2024). However, these methods require training on a per-dataset basis, leading to high computational costs, and they are generally slower than GBDTs for tabular classification tasks.

**Linear Attention for In-Context Learning.** Although linear attention has been reported to underperform in some language modeling tasks (You et al., 2024; Zhang et al., 2024; Qin et al., 2022), recent theoretical work demonstrates its effectiveness in in-context learning scenarios, where it can emulate gradient descent to achieve learning during inference (Ahn et al., 2023).

# B ACCELERATING COMPUTATION THROUGH SAMPLE SELECTION

**Test-Specific Sample Selection.** We conducted additional experiments on three datasets where TabPFN with standard random sample selection underperformed. To enhance efficiency, we employed TabPFN with 1000 nearest-neighbor (KNN) sample selections (instead of 300) and evaluated results based on 100 test samples. Our findings show that sample selection improves ICL performance.

| Dataset | #Classes | #Features | #Instances | TABPFN | | TABFLEX |
|---------|----------|-----------|------------|--------|--------|---------|
| | | | | Random Sample Selection | KNN Sample Selection | |
| SpeedDating | 2 | 120 | 8378 | 0.55 | 0.73 | **0.85** |
| Bioresponse | 2 | 1776 | 3751 | 0.50 | 0.51 | **0.75** |
| nomao | 2 | 118 | 34465 | 0.76 | 0.99 | **0.99** |

Table 5: Results of TabPFN with different test-specific sample selection methods across three datasets.

However, there are significant challenges in using this method for large datasets, primarily due to high computational overhead caused by two factors:

- **Inability to use batch inference**: Since the in-context samples vary for each test instance, we need to recompute the attention outputs for every test sample individually. Our experiments demonstrate that without batch inference, inference times can increase by $1000\times$ or more in practice. For example, with 1000 test samples, our method requires 1000 separate forward passes, compared to batch processing which can classify all of them in a single forward pass.

- **Additional time complexity from sample selection**: Identifying and selecting the nearest samples introduces an extra computational burden, further impacting efficiency.

**Global Sample Selection.** It is also feasible to select important samples from the entire dataset. Methods for selecting significant samples are commonly used in various domains, such as active learning (Settles, 2009; Ren et al., 2021) and addressing subpopulation shifts (Zeng et al., 2022; Hashimoto et al., 2018; Liu et al., 2021). However, these approaches often involve training a model first before selecting key samples (Zeng et al., 2022; Hashimoto et al., 2018; Liu et al., 2021). The key idea is, a model can be trained initially to identify important samples near the decision boundary. However, these approaches introduces significant computational overhead, which contradicts our goal of efficiency. Therefore, we conduct a more simplified sample selection, which perform clustering on samples, and then sample the samples from different clusters for increasing the diversity of dataset, and this is a commonly-known way to help machine learning performance (Gong et al., 2019).

In this experiment, we perform K-means on the training dataset with $k = 10$, and then select 300 samples from each, resulting in total 3000 training samples. The results are presented below. We observe that performance remains largely unchanged.

| Dataset | #Classes | #Features | #Instances | TABPFN | | TABFLEX |
| --- | --- | --- | --- | --- | --- | --- |
| | | | | Random Sample Selection | KNN Sample Selection | |
| airlines | 2 | 7 | 539383 | 0.63 | 0.63 | 0.64 |
| poker-hand | 10 | 10 | 1025009 | 0.72 | 0.71 | 0.84 |

Table 6: Results of TABPFN with different global sample selection methods across two datasets.

## C  EVALUATING OTHER ATTENTION MECHANISMS

In addition to the broad categories of all linear RNN variant models we studied in this paper, we also consider another mechanism that enjoys linear complexity: sliding window attention (Beltagy et al., 2020). We show that TABFLEX achieves significantly better performance.

| Method | #Class | #Features | #Instances | Sliding Window | Linear (Ours) |
| --- | --- | --- | --- | --- | --- |
| Poker-Hand | 10 | 10 | 1,025,009 | 0.48 | **0.84** |
| Airlines | 2 | 7 | 539,383 | 0.48 | **0.64** |
| Higgs | 2 | 28 | 98,050 | 0.39 | **0.76** |

Table 7: Performance comparison of TABFLEX with Sliding Window attention.

## D  COMPUTATION ANALYSIS OF VARIOUS ATTENTION MECHANISM

In this section, we provide a computational analysis of various attention mechanisms, comparing standard attention, FlashAttention (specifically FlashAttention-I (Dao et al., 2022)), causal Flash-LinearAttention (referred to as FlashLinearAttention in Yang et al. (2024)), and non-causal linear attention. To clarify, FlashLinearAttention is designed to reduce HBM access specifically for causal linear attention. For notational simplicity, we use the term "linear attention" to refer to non-causal linear attention.

---

**Algorithm 2:** Causal FlashLinearAttention Implementation (Yang et al., 2024)

**Input:** Matrices $\boldsymbol{Q}, \boldsymbol{K}, \boldsymbol{V} \in \mathbb{R}^{N \times D}$ in HBM, on-chip SRAM of size $M$

1   Set block size $B$;

2   Initialize $\boldsymbol{O} = (0)_{N \times D} \in \mathbb{R}^{N \times D}$ in HBM;

3   Divide $\boldsymbol{Q}$ into $T = \lceil \frac{N}{B} \rceil$ blocks $\boldsymbol{Q}_1, \ldots, \boldsymbol{Q}_T$ of size $B \times D$ each, and divide $\boldsymbol{K}, \boldsymbol{V}$ into
    $T = \lceil \frac{N}{B} \rceil$ blocks $\boldsymbol{K}_1, \ldots, \boldsymbol{K}_T$ and $\boldsymbol{V}_1 \ldots \boldsymbol{V}_T$ of size $B \times D$ each;

4   Divide $\boldsymbol{O}$ into $T$ blocks $\boldsymbol{O}_1, \ldots, \boldsymbol{O}_T$ of size $B \times D$ each;

5   On on-chip SRAM, construct causal mask, $\boldsymbol{M} \in \mathbb{R}^{B \times B}$;

6   On SRAM, initialize $\boldsymbol{S} = (0)_{D \times D} \in \mathbb{R}^{D \times D}$;

7   **for** $1 \leq j \leq T$ **do**

8      Load $\boldsymbol{K}_j, \boldsymbol{V}_j, \boldsymbol{Q}_j, \boldsymbol{O}_j$ from HBM to on-chip SRAM;

9      Write $\boldsymbol{O}_j \leftarrow \boldsymbol{Q}_j \boldsymbol{S} + ((\boldsymbol{Q}_j \boldsymbol{K}_j^\top) \odot \boldsymbol{M}) \cdot \boldsymbol{V}_j$ to HBM;

10     On chip, compute $\boldsymbol{S} \leftarrow \boldsymbol{S} + \boldsymbol{K}_j^\top \boldsymbol{V}_j$;

11 **end**

**Output:** $\boldsymbol{O}$

---

We evaluate these mechanisms based on their High Bandwidth Memory (HBM) access, memory requirements, and floating-point operations per second (FLOPS) when computing attention outputs given query, key, and value inputs. While Dao et al. (2022) have provided computations for standard attention and FlashAttention, we focus our analysis on causal FlashLinearAttention (detailed in Alg. 2) and HBM-efficient non-causal linear attention (developed by us and detailed in Alg. 3) in Sec. D.1. In practice, we employ a simplified PyTorch implementation of linear attention and demonstrate its efficiency, as it only causes marginal increases in HBM access and memory usage as we demonstrate in Sec. D.2. Furthermore, we present visualizations in Sec. D.2 that illustrate

---

**Algorithm 3:** HBM-Efficient Implementation of Linear Attention

---

**Input:** Matrices $\boldsymbol{Q}, \boldsymbol{K}, \boldsymbol{V} \in \mathbb{R}^{N \times D}$ in HBM, on-chip SRAM of size $M$

1 Set block size $B$;

2 Initialize $\boldsymbol{O} = (0)_{N \times D} \in \mathbb{R}^{N \times D}$ in HBM;

3 Divide $\boldsymbol{Q}$ into $T = \lceil \frac{N}{B} \rceil$ blocks $\boldsymbol{Q}_1, \ldots, \boldsymbol{Q}_T$ of size $B \times D$ each, and divide $\boldsymbol{K}, \boldsymbol{V}$ into
$T = \lceil \frac{N}{B} \rceil$ blocks $\boldsymbol{K}_1, \ldots, \boldsymbol{K}_T$ and $\boldsymbol{V}_1, \ldots, \boldsymbol{V}_T$ of size $B \times D$ each;

4 Divide $\boldsymbol{O}$ into $T$ blocks $\boldsymbol{O}_1, \ldots, \boldsymbol{O}_T$ of size $B \times D$ each;

5 On on-chip SRAM, initialize $\boldsymbol{S} = (0)_{D \times D} \in \mathbb{R}^{D \times D}$;

6 **for** $1 \leq i \leq T$ **do**

7     Load $\boldsymbol{K}_i, \boldsymbol{V}_i$;

8     On chip, compute $\boldsymbol{S} \leftarrow \boldsymbol{S} + \boldsymbol{K}_i^\top \boldsymbol{V}_i$;

9 **for** $1 \leq j \leq T$ **do**

10     Load $\boldsymbol{Q}_j, \boldsymbol{O}_j$;

11     Write $\boldsymbol{O}_j \leftarrow \boldsymbol{Q}_j \boldsymbol{S}$ to HBM;

**Output:** $\boldsymbol{O}$

---

### D.1 HBM-Efficient Linear Attention

In this section, we analyze the number of HBM accesses, HBM memory, and FLOPS required by FlashLinearAttention (Alg. 2) and linear attention (Alg. 3).

**Lemma 2.** *Let $\boldsymbol{Q}, \boldsymbol{K}, \boldsymbol{V} \in \mathbb{R}^{N \times D}$ represent the query, key, and value matrices for a single attention head, where $N$ is the sequence length and $D$ is the embedding size. Both FlashLinearAttention (Alg. 2) and linear attention (Alg. 3) require $5ND$ HBM accesses to compute the attention output.*

*Proof of Lemma 2.* For causal FlashLinearAttention (Alg. 2):

- Line 8: Loading $\boldsymbol{K}_j, \boldsymbol{V}_j, \boldsymbol{Q}_j, \boldsymbol{O}_j$ necessitates $4BD$ HBM accesses.

- Line 9: Writing $\boldsymbol{O}_j$ requires $BD$ HBM accesses.

These operations are executed $T$ times, where $T = \lceil \frac{N}{B} \rceil$. Thus, the total HBM accesses are:

$$5BD \cdot T = 5BD \cdot \lceil \frac{N}{B} \rceil = 5ND.$$

For non-causal linear attention (Alg. 3):

- Line 7: Loading $\boldsymbol{K}_i, \boldsymbol{V}_i$ requires $2BD$ HBM accesses.

- Line 10: Loading $\boldsymbol{Q}_j, \boldsymbol{O}_j$ demands $2BD$ HBM accesses.

- Line 11: Writing $\boldsymbol{O}_j$ necessitates $BD$ HBM accesses.

These operations are also repeated $T$ times, where $T = \lceil \frac{N}{B} \rceil$. Consequently, the total HBM accesses are:

$$5BD \cdot T = 5BD \cdot \lceil \frac{N}{B} \rceil = 5ND.$$

Therefore, we conclude that both causal FlashLinearAttention and non-causal linear attention require $5ND$ HBM accesses to compute the attention output. $\qquad\square$

**Lemma 3.** *Let $\boldsymbol{Q}, \boldsymbol{K}, \boldsymbol{V} \in \mathbb{R}^{N \times D}$ represent the query, key, and value matrices for a single attention head, where $N$ is the sequence length and $D$ is the embedding size. Both FlashLinearAttention (Alg. 2) and linear attention (Alg. 3) require $4ND$ HBM memory to compute the attention output.*

*Proof of Lemma 3.* For both algorithms:

- Storing $\boldsymbol{Q}, \boldsymbol{K}, \boldsymbol{V}$ requires $3ND$ memory.

- Storing $\boldsymbol{O}$ requires $ND$ memory.

Total HBM memory usage: $4ND$. $\qquad\square$

**Lemma 4.** *Let $\boldsymbol{Q}, \boldsymbol{K}, \boldsymbol{V} \in \mathbb{R}^{N \times D}$ represent the query, key, and value matrices for a single attention head, where $N$ is the sequence length and $D$ is the embedding size. Both FlashLinearAttention (Alg. 2) and linear attention (Alg. 3) require $O(ND^2)$ FLOPS to compute the attention output.*

*Proof of Lemma 4.* For causal FlashLinearAttention (Alg. 2):

- Computing $(\boldsymbol{Q}_j \boldsymbol{K}_j^\top) \odot \mathbf{M}$ requires $B^2(2D-1) + B^2$ FLOPs.

- The result of step 1 multiplied by $\boldsymbol{V}_j$ requires $B^2(2D-1) + BD(2B-1)$ FLOPs.

- Computing $\boldsymbol{Q}_j \boldsymbol{S}$ requires $B \cdot D(2D-1)$ FLOPs.

- Computing $\boldsymbol{K}_j^\top \boldsymbol{V}_j$ (line 10) requires $(2B-1) \cdot D^2$ FLOPs.

The total number of FLOPs for one iteration is:
$$B^2(2D-1) + B^2 + B^2(2D-1) + BD(2B-1) + B \cdot D(2D-1) + (2B-1) \cdot D^2$$
$$= 4B^2D - BD + 4BD^2 - D^2.$$

These operations are repeated $T = \lceil \frac{N}{B} \rceil$ times. The total number of FLOPs is:
$$(4B^2D - BD + 4BD^2 - D^2) \cdot T = O(ND^2).$$

For non-causal linear attention (Alg. 3):

- Computing $\boldsymbol{K}_i^\top \boldsymbol{V}_i$ (line 8) requires $D^2(2B-1)$ FLOPs.

- Computing $\boldsymbol{Q}_j \boldsymbol{S}$ (line 11) requires $(2D-1)BD$ FLOPs.

These operations are repeated $T = \lceil \frac{N}{B} \rceil$ times. The total number of FLOPs is:
$$(2BD^2 - D^2 + 2BD^2 - BD) \cdot T = O(ND^2).$$

Thus, we conclude that both algorithms require $O(ND^2)$ FLOPs to compute the attention output.
$\qquad\square$

### D.2 SIMPLIFIED PYTORCH IMPLEMENTATION OF LINEAR ATTENTION

In our implementation, we adopt a straightforward PyTorch approach to linear attention rather than an HBM-efficient method. We employ the concise two-line implementation presented in Listing 1. In the following lemma, we demonstrate that this straightforward implementation only incurs a marginal increase in HBM accesses and HBM memory usage.

```python
def linear_attn(q, k, v):
    """
    q: (batch, heads, seq_q, dim_qk)
    k: (batch, heads, seq_kv, dim_qk)
    v: (batch, heads, seq_kv, dim_v)
    """
    kv = torch.einsum("bhnd,bhnm->bhdm", k, v)
    o = torch.einsum("bhld,bhdm->bhlm", q, kv)
    return o.contiguous()
```

Listing 1: Straightforward PyTorch implementation of linear attention (Katharopoulos et al., 2020).

**Theorem 1.** *Let $Q, K, V \in \mathbb{R}^{N \times D}$ represent the query, key, and value matrices for a single attention head, where $N$ is the sequence length and $D$ is the embedding size. Both causal FlashLinearAttention (Alg. 2) and non-causal linear attention (Listing 1) require $O(ND)$ HBM accesses, $O(ND)$ HBM memory, and $O(ND^2)$ FLOPS to compute the attention output.*

*Proof.* Let us consider the implementation in Listing 1 and compare it to Alg. 3. PyTorch's optimized tensor computation ensures efficiency, with the primary distinction between Listing 1 and Alg. 3 being the storage of kv in the former, which is equivalent to $S \in \mathbb{R}^{D \times D}$ in Alg. 3. This results in the following changes:

- HBM Accesses: By Lemma 2, Alg. 3 requires $5ND$ HBM accesses. Due to the additional write and load operations for $S \in \mathbb{R}^{D \times D}$, Listing 1 requires $5ND + 2D^2$ HBM accesses.

- HBM Memory Usage: By Lemma 3, Alg. 3 requires $4ND$ HBM memory usage. Due to the additional storage requirements for $S \in \mathbb{R}^{D \times D}$, Listing 1 requires $4ND + D^2$ HBM memory usage.

The number of FLOPS remains unaffected. The analysis above, in conjunction with Lemmas 2, 3, and 4, yields the desired outcome. $\square$

In Table 8, we summarize the #HBM access, HBM memory, and FLOPS required by standard attention (with naive PyTorch implementation), FlashAttention-I, FlashLinearAttention (causal), and linear attention with both implementations.

| | **Standard Attention** | **FlashAttention** (Dao et al., 2022) | **FlashLinearAttention** (Yang et al., 2024) | **Linear Attention** Alg. 3 | Listing 1 |
|---|---|---|---|---|---|
| # HBM access | $4N^2 + 4ND$ | $\frac{12N^2D^2}{M} + \frac{16N^2D}{M} + 2ND$ | $5ND$ | $5ND$ | $5ND + 2D^2$ |
| Memory | $2N^2 + 4ND$ | $2N + 4ND$ | $4ND$ | $4ND$ | $4ND + D^2$ |
| FLOPS | $O(N^2D)$ | $O(N^2D)$ | $O(ND^2)$ | $O(ND^2)$ | $O(ND^2)$ |

Table 8: **Comparison of memory and computational costs across different attention mechanisms.** FlashAttention improves the speed of standard attention by optimizing # HBM access. Flash causal linear attention takes a similar approach, achieving linear # HBM access. However, we show that non-causal linear attention already achieves linear # HBM access, matching the efficiency of flash causal linear attention without requiring any additional optimization on # HBM access.

Subsequently, we visualize the empirical execution time and CUDA memory utilization of FlashAttention-2, FlashLinearAttention, and linear attention in Fig. 5a and Fig. 5b, respectively. We vary the head dimension $\in \{32, 64, 128, 256\}$, the number of heads $\in \{2, 4, 8, 16\}$, and the sequence length $\in \{2^4, 2^5, \ldots, 2^{15}\}$. We focus on the self-attention case, randomly generating input (serving as key, query, and values) with a batch size of 10, and replicate the experiment 5 times. The final values presented are aggregated across these 5 simulations. Notably, we were unable to obtain results for FlashLinearAttention in two configurations: (1) head dimension 256 with 8 heads, and (2) head dimension 256 with 16 heads, due to illegal memory access error incurred by the PyTorch package fla (Yang et al., 2024). Our observations from the figures indicate that both runtime and CUDA memory usage of FlashLinearAttention and linear attention exhibit linear growth with respect to sequence length, aligning with the predictions of Theorem 1.

# E    DETAILS OF TABFLEX

In this section, we elucidate the finer details of TABFLEX, encompassing our model training details and validation dataset selection process.

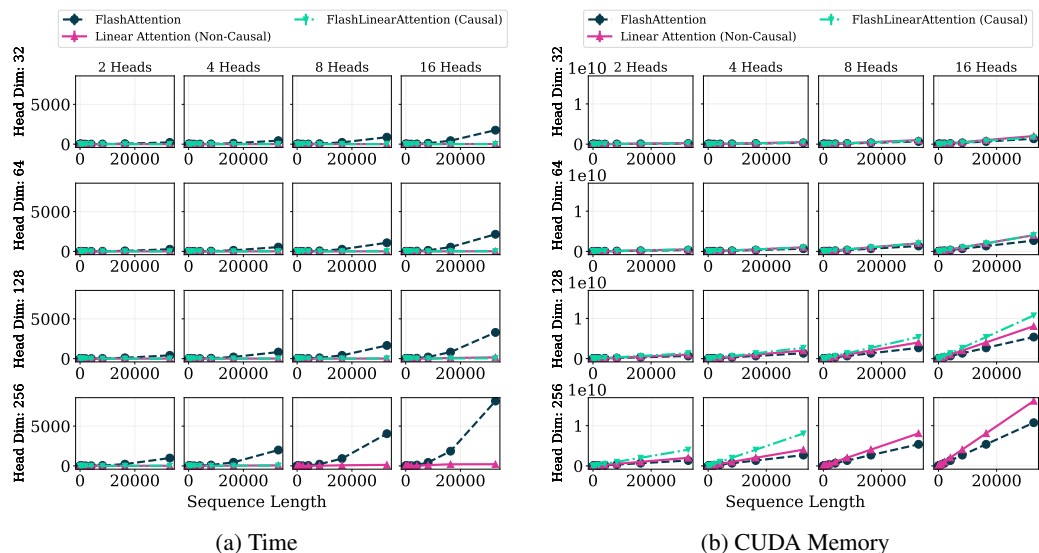

(a) Time         (b) CUDA Memory

Figure 5: Time and CUDA memory usage comparison of FlashAttention-2 (Dao, 2024), causal FlashLinearAttention (Yang et al., 2024), and linear attention (Katharopoulos et al., 2020) (implemented as in Listing 1). Results for FlashLinearAttention in two configurations: (1) head dimension 256 with 8 heads, and (2) head dimension 256 with 16 heads are missing, due to illegal memory access error incurred by the PyTorch package `fla` (Yang et al., 2024).

### E.1 MODEL TRAINING

We implement linear attention with the feature function $\texttt{elu}(\cdot) + 1$, adhering to the default implementation proposed by Katharopoulos et al. (2020). Unless otherwise specified, we adopt the training setup of TABPFN for TABFLEX-S100, TABFLEX-L100, and TABFLEX-H1K. Each model is trained on a single Nvidia A100 80GB PCIe GPU.

| Hyperparameters | Batch Size | Epoch | Learning Rate | #Steps/epoch |
|---|---|---|---|---|
| TABFLEX-S100 | 1210 | 8 | 3e-5 | 8192 |
| TABFLEX-L100 | 110 | 4 | 3e-5 | 8192 |
| TABFLEX-H1K | 1410 | 4 | 3e-5 | 1024 |

Table 9: **Hyperparameters used for training TABFLEX models.** The number of steps per epoch indicates the quantity of synthetic datasets generated and used for training within each epoch.

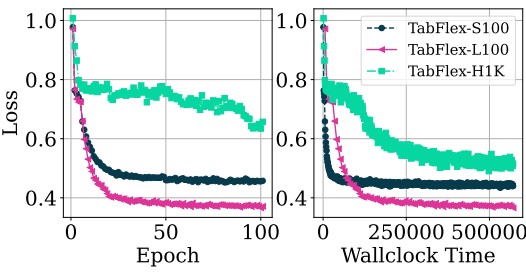

Figure 6: Visualization of training loss for TABFLEX models as a function of epoch and wall-clock time.

Table 9 summarizes the hyperparameters selected for training TABFLEX-S100, TABFLEX-L100, and TABFLEX-H1K. For all three methods, we utilize the same embedding size of 512, consistent with TABPFN. We extend the feature capacity by modifying the first linear layer, which projects the features into embeddings – specifically, we increase the number of neurons responsible for receiving the features.

The training loss curves are illustrated in Fig. 6. We observe that as the number of features and the length of training dataset sequences increase, the training process becomes more time-consuming. In fact, training a robust TABFLEX-H1K model requires more than three weeks.

## E.2 Validation Datasets

We select the validation datasets from the OpenML AutoML Benchmark (Feurer et al., 2021) by choosing 10 datasets from each of the following sample size intervals: $[0.1K, 1K)$, $[1K, 10K)$, and $[10K, 100K)$. To ensure diversity in the validation set, we also vary the number of classes and features within each interval. The details of all datasets used in validation are summarized in Table 10.

| OpenML did | Dataset | #Features | #Instances | #Classes |
|---|---|---|---|---|
| 279 | meta-stream-intervals.arff | 75 | 45164 | 11 |
| 311 | oil-spill | 50 | 937 | 2 |
| 742 | fri-c4-500-100 | 101 | 500 | 2 |
| 825 | boston-corrected | 21 | 506 | 2 |
| 833 | bank32nh | 33 | 8192 | 2 |
| 841 | stock | 10 | 950 | 2 |
| 920 | fri-c2-500-50 | 51 | 500 | 2 |
| 940 | water-treatment | 37 | 527 | 2 |
| 981 | kdd-internet-usage | 69 | 10108 | 2 |
| 1039 | hiva-agnostic | 1618 | 4229 | 2 |
| 1491 | one-hundred-plants-margin | 65 | 1600 | 100 |
| 1492 | one-hundred-plants-shape | 65 | 1600 | 100 |
| 1503 | spoken-arabic-digit | 15 | 263256 | 10 |
| 1515 | micro-mass | 1301 | 571 | 20 |
| 1536 | volcanoes-b6 | 4 | 10130 | 5 |
| 1541 | volcanoes-d4 | 4 | 8654 | 5 |
| 1549 | autoUniv-au6-750 | 41 | 750 | 8 |
| 40645 | GAMETES-Epistasis-2-Way-1000atts-0.4H-EDM-1-EDM-1-1 | 1001 | 1600 | 2 |
| 40672 | fars | 30 | 100968 | 8 |
| 40677 | led24 | 25 | 3200 | 10 |
| 40693 | xd6 | 10 | 973 | 2 |
| 40705 | tokyo1 | 45 | 959 | 2 |
| 40922 | Run-or-walk-information | 7 | 88588 | 2 |
| 40985 | tamilnadu-electricity | 4 | 45781 | 20 |
| 41082 | USPS | 257 | 9298 | 10 |
| 41144 | madeline | 260 | 3140 | 2 |
| 41986 | GTSRB-HOG01 | 1569 | 51839 | 43 |
| 41988 | GTSRB-HOG02 | 1569 | 51839 | 43 |
| 41989 | GTSRB-HOG03 | 2917 | 51839 | 43 |
| 41990 | GTSRB-HueHist | 257 | 51839 | 43 |
| 41991 | Kuzushiji-49 | 785 | 270912 | 49 |
| 42193 | compas-two-years | 14 | 5278 | 2 |
| 42206 | porto-seguro | 38 | 595212 | 2 |
| 42343 | KDD98 | 478 | 82318 | 2 |

Table 10: Characteristics of datasets in our diverse validation set.

# F Supplementary Experimental Details and Results

In this section, we present the details of the test datasets and additional experiment results.

## F.1 TabZilla Datasets

The results of our experiments on TabZilla-related datasets are reported in Table 1, 2, and 3. (McElfresh et al., 2023) presents the details of the datasets used in their hard benchmark (Table 3) in Table 4 of their paper. We provide the specifications of the datasets used for our evaluation in Table 1 and Table 2 in Table 11 and Table 12, respectively.

| Dataset | D | N | C | Dataset | D | N | C | Dataset | D | N | C |
|---|---|---|---|---|---|---|---|---|---|---|---|
| cmc | 9 | 1473 | 3 | socmob | 5 | 1156 | 1 | adult-census | 14 | 32561 | 2 |
| kc1 | 21 | 2109 | 1 | vehicle | 18 | 846 | 4 | breast-cancer | 9 | 286 | 2 |
| kc2 | 21 | 522 | 1 | heart-h | 13 | 294 | 1 | mfeat-factors | 216 | 2000 | 10 |
| pc3 | 37 | 1563 | 1 | jasmine | 144 | 2984 | 1 | mfeat-zernike | 47 | 2000 | 10 |
| pc4 | 37 | 1458 | 1 | phoneme | 5 | 5404 | 1 | dresses-sales | 12 | 500 | 2 |
| pc1 | 21 | 1109 | 1 | semeion | 256 | 1593 | 10 | mfeat-fourier | 76 | 2000 | 10 |
| cjs | 33 | 2796 | 6 | heart-c | 13 | 303 | 1 | balance-scale | 4 | 625 | 3 |
| car | 6 | 1728 | 4 | kr-vs-kp | 36 | 3196 | 1 | bank-marketing | 16 | 45211 | 2 |
| tae | 5 | 151 | 3 | spambase | 57 | 4601 | 1 | car-evaluation | 21 | 1728 | 4 |
| jm1 | 21 | 10885 | 1 | satimage | 36 | 6430 | 6 | cylinder-bands | 37 | 540 | 2 |
| dna | 180 | 3186 | 3 | mushroom | 22 | 8124 | 1 | mfeat-karhunen | 64 | 2000 | 10 |
| musk | 167 | 6598 | 1 | diabetes | 8 | 768 | 1 | credit-approval | 15 | 690 | 2 |
| wdbc | 30 | 569 | 1 | rabe_266 | 2 | 120 | 1 | ozone-level-8hr | 72 | 2534 | 2 |
| wilt | 5 | 4839 | 1 | breast-w | 9 | 699 | 1 | analcatdata_dmft | 4 | 797 | 6 |
| ilpd | 10 | 583 | 1 | elevators | 18 | 16599 | 1 | monks-problems-2 | 6 | 601 | 2 |
| sick | 28 | 3772 | 1 | Satellite | 36 | 5100 | 1 | cardiotocography | 35 | 2126 | 10 |
| iris | 4 | 150 | 3 | fertility | 9 | 100 | 1 | PhishingWebsites | 30 | 11055 | 2 |
| lymph | 18 | 148 | 4 | ionosphere | 34 | 351 | 1 | synthetic_control | 60 | 600 | 6 |
| churn | 20 | 5000 | 1 | transplant | 3 | 131 | 1 | steel-plates-fault | 27 | 1941 | 7 |
| colic | 22 | 368 | 1 | eucalyptus | 19 | 736 | 5 | mfeat-morphological | 6 | 2000 | 10 |
| ecoli | 7 | 336 | 8 | Australian | 14 | 690 | 1 | acute-inflammations | 6 | 120 | 2 |
| autos | 25 | 205 | 6 | hayes-roth | 4 | 160 | 3 | analcatdata_boxing1 | 3 | 120 | 2 |
| scene | 299 | 2407 | 1 | dermatology | 34 | 366 | 6 | analcatdata_chlamydia | 3 | 100 | 2 |
| profb | 9 | 672 | 1 | MiceProtein | 77 | 1080 | 8 | wall-robot-navigation | 24 | 5456 | 4 |
| colic | 26 | 368 | 1 | SpeedDating | 120 | 8378 | 1 | visualizing_livestock | 2 | 130 | 2 |
| labor | 16 | 57 | 1 | tic-tac-toe | 9 | 958 | 1 | Click_prediction_small | 1 | 39948 | 2 |
| irish | 5 | 500 | 1 | hill-valley | 100 | 1212 | 1 | analcatdata_authorship | 70 | 841 | 4 |
| glass | 9 | 214 | 6 | page-blocks | 10 | 5473 | 5 | banknote-authentication | 4 | 1372 | 2 |
| yeast | 8 | 1269 | 4 | lung-cancer | 56 | 32 | 3 | LED-display-domain-7digit | 7 | 500 | 10 |
| sonar | 60 | 208 | 1 | qsar-biodeg | 41 | 1055 | 1 | visualizing-environmental | 3 | 111 | 2 |
| splice | 60 | 3190 | 3 | fri_c3_100_5 | 5 | 100 | 1 | postoperative-patient-data | 8 | 88 | 2 |
| libras | 104 | 360 | 10 | ada_agnostic | 48 | 4562 | 1 | blood-transfusion-service-center | 4 | 748 | 2 |
| anneal | 38 | 898 | 5 | fri_c0_100_5 | 5 | 100 | 1 | | | | |

Table 11: Datasets utilized in the evaluation presented in Table 1. Here $D$, $N$, and $C$ denote the number of features, instances, and classes, respectively.

| Dataset | #Features | #Instances | #Classes |
|---|---|---|---|
| Australian | 14 | 690 | 2 |
| LED-display-domain-7digit | 7 | 500 | 10 |
| MiceProtein | 77 | 1080 | 8 |
| acute-inflammations | 6 | 120 | 2 |
| analcatdata_authorship | 70 | 841 | 4 |
| analcatdata_boxing1 | 3 | 120 | 2 |
| analcatdata_chlamydia | 3 | 100 | 2 |
| analcatdata_dmft | 4 | 797 | 6 |
| anneal | 38 | 898 | 5 |
| autos | 25 | 205 | 6 |
| balance-scale | 4 | 625 | 3 |
| blood-transfusion-service-center | 4 | 748 | 2 |
| blood-transfusion-service-center | 4 | 748 | 2 |
| breast-cancer | 9 | 286 | 2 |
| breast-w | 9 | 699 | 2 |
| colic | 26 | 368 | 2 |
| colic | 22 | 368 | 2 |
| credit-approval | 15 | 690 | 2 |
| cylinder-bands | 37 | 540 | 2 |
| dermatology | 34 | 366 | 6 |
| diabetes | 8 | 768 | 2 |
| dresses-sales | 12 | 500 | 2 |
| ecoli | 7 | 336 | 8 |
| eucalyptus | 19 | 736 | 5 |
| fertility | 9 | 100 | 2 |
| fri_c0_100_5 | 5 | 100 | 2 |
| fri_c3_100_5 | 5 | 100 | 2 |
| glass | 9 | 214 | 6 |
| hayes-roth | 4 | 160 | 3 |
| heart-c | 13 | 303 | 2 |
| heart-h | 13 | 294 | 2 |
| hill-valley | 100 | 1212 | 2 |
| ilpd | 10 | 583 | 2 |
| ionosphere | 34 | 351 | 2 |
| iris | 4 | 150 | 3 |
| irish | 5 | 500 | 2 |
| kc2 | 21 | 522 | 2 |
| labor | 16 | 57 | 2 |
| lung-cancer | 56 | 32 | 3 |
| lymph | 18 | 148 | 4 |
| monks-problems-2 | 6 | 601 | 2 |
| pc1 | 21 | 1109 | 2 |
| postoperative-patient-data | 8 | 88 | 2 |
| profb | 9 | 672 | 2 |
| qsar-biodeg | 41 | 1055 | 2 |
| rabe_266 | 2 | 120 | 2 |
| socmob | 5 | 1156 | 2 |
| sonar | 60 | 208 | 2 |
| synthetic_control | 60 | 600 | 6 |
| tae | 5 | 151 | 3 |
| tic-tac-toe | 9 | 958 | 2 |
| transplant | 3 | 131 | 2 |
| vehicle | 18 | 846 | 4 |
| visualizing_environmental | 3 | 111 | 2 |
| visualizing_livestock | 2 | 130 | 2 |
| wdbc | 30 | 569 | 2 |
| yeast | 8 | 1269 | 4 |

Table 12: Datasets utilized in the evaluation presented in Table 2.

## F.2 EVALUATION ON ADDITIONAL DATASETS

In this section, we provide additional evaluation of TABFLEX on eight large datasets randomly selected from OpenML-CC18 Benchmarks (Bischl et al., 2019), after excluding the datasets contained in TabZilla's evaluation. As shown in Table 13, TABFLEX consistently outperforms TABPFN in terms of speed and achieves superior performance on the majority of the datasets.

| Dataset | #Features | #Instances | #Classes | Mean AUC | | Mean Time (seconds) | |
|---------|-----------|------------|----------|----------|----------|----------|----------|
| | | | | TABPFN | TABFLEX | TABPFN | TABFLEX |
| kick | 33 | 72983 | 2 | 0.663 | **0.684** | 13.330 | **3.096** |
| Click-prediction-small-1220 | 10 | 39948 | 2 | 0.652 | **0.659** | 3.663 | **0.887** |
| house-8L | 9 | 22784 | 2 | **0.947** | 0.945 | 1.383 | **0.536** |
| okcupid-stem | 20 | 50789 | 3 | 0.825 | **0.828** | 6.152 | **1.511** |
| volcanoes-b1 | 4 | 10176 | 5 | 0.660 | **0.663** | 0.349 | **0.202** |
| volcanoes-b2 | 4 | 10668 | 5 | 0.651 | **0.652** | 0.375 | **0.217** |
| kdd-internet-usage | 69 | 10108 | 2 | **0.932** | **0.932** | 1.021 | **0.851** |
| BNG(tic-tac-toe) | 10 | 39366 | 2 | **0.836** | 0.835 | 3.626 | **1.111** |

Table 13: **Performance comparison between TABPFN and TABFLEX on an additional large dataset.** We observe that TABFLEX is consistently faster than TABPFN and outperforms it on the majority of the datasets.

## F.3 ADDITIONAL COMPARISON WITH TUNETABLES

As mentioned in Sec. 6, the results of TuneTables presented in Table 14 of our main experiments use `TuneTablesClassifier`. However, we note that the original paper reported results after 30 iterations of hyperparameter tuning. They also applied this process to TABPFN, using a different subset of datasets as training samples at each iteration. In Table 14, we compare the performance of TABFLEX without any hyperparameter tuning to the results reported in their paper. TABFLEX remains competitive, particularly when the number of samples is limited. While TuneTables tends to perform better with larger sample sizes due to its ability to update model parameters based on training data, TABFLEX maintains comparable performance while being significantly faster.

| Dataset | Size | TABPFN | | TuneTables | | TABFLEX | |
|---|---|---|---|---|---|---|---|
| | | Acc. | Runtime (sec.) | Acc. | Runtime (sec.) | Acc. | Runtime (sec.) |
| breast-cancer | 286 | .765 | 29 | .770 | 65 | **.793** | 1 |
| heart-c | 303 | .848 | 40 | **.903** | 66 | **.903** | 0 |
| ecoli | 336 | .848 | 30 | .843 | 66 | .882 | 0 |
| colic | 368 | .856 | 39 | **.892** | 66 | **.892** | 0 |
| dresses-sales | 500 | .578 | 41 | **.580** | 122 | **.580** | 0 |
| cylinder-bands | 540 | .800 | 41 | **.846** | 82 | .796 | 0 |
| climate | 540 | .959 | 59 | .951 | 97 | **.963** | 0 |
| balance-scale | 625 | .990 | 29 | .995 | 55 | **1.000** | 0 |
| blood-transfusion | 748 | .801 | 25 | .782 | 56 | **.840** | 0 |
| cmc | 1473 | .554 | 91 | .556 | 109 | **.605** | 0 |
| kc-1 | 2109 | .862 | 168 | .856 | 187 | **.867** | 0 |
| bioresponse | 3151 | .797 | 638 | **.798** | 3012 | .720 | 13 |
| christine | 5418 | .742 | 666 | **.755** | 3920 | .721 | 11 |
| robert | 10000 | .250 | 964 | **.414** | 2397 | .333 | 17 |
| dilbert | 10000 | .922 | 761 | **.992** | 3749 | .802 | 17 |
| har | 10299 | .936 | 370 | **.981** | 2657 | .918 | 9 |
| eeg-eye-state | 14980 | .940 | 178 | **.986** | 1929 | .837 | 1 |
| elevators | 16599 | .902 | 186 | .902 | 1297 | **.907** | 1 |
| riccardo | 20000 | .922 | 1395 | **.995** | 5247 | .773 | 31 |
| volkert | 58310 | .567 | 459 | **.693** | 6331 | .561 | 12 |
| higgs | 67557 | .671 | 931 | **.714** | 4084 | .691 | 1 |
| connect-4 | 98050 | .668 | 931 | **.817** | 5395 | .692 | 1 |
| BNG (vote) | 131072 | .968 | 1976 | **.974** | 2493 | **.974** | 1 |
| albert | 425240 | .642 | 2363 | **.658** | 17518 | .637 | 1 |
| airlines | 539383 | .600 | 2602 | **.653** | 44434 | .597 | 2 |
| BNG (labor) | 1000000 | .937 | 5518 | **.967** | 7717 | .950 | 8 |
| agrawall | 1000000 | .948 | 5158 | **.950** | 45504 | .948 | 3 |
| poker-hand | 1025009 | .531 | 2423 | **1.000** | 10471 | .542 | 15 |
| click-prediction-small | 1997410 | .833 | 10421 | **.837** | 33148 | .833 | 5 |

Table 14: Accuracy comparison of TABPFN, TuneTables, and TABFLEX on test datasets from Feuer et al. (2024). Results for TABPFN and TuneTables are directly sourced from Feuer et al. (2024), where hyperparameter tuning was performed 30 times for both methods. For TABPFN, hyperparameters determine the subset of the dataset used in ICL. TABFLEX results are reported without hyperparameter tuning.

