# OpenReview forum: "TabFlex: Scaling Tabular Learning to Millions with Linear Attention"
_ICLR.cc/2025/Conference — Submitted to ICLR 2025_

### Official Review · Reviewer_VkbL · 2024-10-28

**Soundness:** 3
**Presentation:** 3
**Contribution:** 1
**Rating:** 5
**Confidence:** 4

**Summary:**

The authors introduce TabFLEX. TabFLEX uses the straightforward PyTorch implementation of linear attention to achieve considerable speedup compared to TabPFN. The authors also train new PFN checkpoints designed to handle larger contexts, taking advantage of linear attention's more efficient inference to achieve faster wall-clock performance and improved utility compared to TabPFN and other baselines.

**Strengths:**

* The authors train and release new PFN checkpoints with a broader range of compatibility and utility, compared to TabPFN.
* The writing is clear.
* The authors provide many experiments demonstrating the efficacy of their method.
* The authors release code to reproduce their work.

**Weaknesses:**

MAJOR

* This paper's core technical contribution is, as the authors put it, utilizing the straightforward PyTorch implementation of linear attention to replace softmax attention in TabPFN [https://pytorch.org/docs/stable/generated/torch.nn.functional.scaled_dot_product_attention.html]. Why does TabPFN not already use linear attention? As of PyTorch 2.0, this can often happen by default, with no changes required to the code base. Even if some minor alterations to the code were necessary, it's not clear why this simple engineering task is worthy of a standalone paper.
* Because switching to linear attention is such a straightforward process and does not require model retraining, it would be easy to transition *all* of the transformer-based methods described in the paper; this would be a fairer comparison, since the authors did not contribute a new learning algorithm. If this was done, I would expect most baselines to improve substantially, including TabPFN.
* Given that the choice of linear attention method is a main technical contribution, it would have been very helpful to see comparisons to more baselines than just a SSM; there are many variants of linear attention available in PyTorch [https://pytorch.org/blog/flexattention/].
* The authors' finding that the inherent causality of SSMs impedes ICL performance compared to non-causal mechanisms is quite interesting, but the experimental results don't totally support this argument. Fig. 2(a) caption states, "The non-causal masked model shows better sample utilization and accuracy as the number of samples grows. In contrast, the causal model’s accuracy declines." However, this is not the actual effect shown; accuracy improves *up to a point* (appears to be around 500 samples), then declines. The authors' argument about causality doesn't explain why there should be an inflection point well past the zero-sample mark, only after which performance declines.

MINOR

* The layout of Fig. 4 contains some overlapping text.

**Questions:**

* Why did the authors not consider more variants of linear attention?

---

> ### Author Response · Authors · 2024-11-26
>
> We thank the reviewer for acknowledging that our paper provides many experiments, our method is broader in range and utility, and we release code for reproducibility. However, we believe there have been some *major misunderstanding* of certain aspects of our work, and we are happy to clarify these points for you.
> ***
> > Q: This paper's core technical contribution is, as the authors put it, utilizing the straightforward PyTorch implementation of linear attention to replace softmax attention in TabPFN [https://pytorch.org/docs/stable/generated/torch.nn.functional.scaled_dot_product_attention.html]. Why does TabPFN not already use linear attention? As of PyTorch 2.0, this can often happen by default, with no changes required to the code base. Even if some minor alterations to the code were necessary, it's not clear why this simple engineering task is worthy of a standalone paper.
>
> Thank you for your question. We believe there may be some confusion between **flash attention** (as per the link you shared) and **linear attention**, which are fundamentally different in design and complexity. To clarify: the link you provided refers to flash attention, an I/O-aware optimization of softmax attention that still has quadratic complexity. In contrast, our work leverages linear attention, which reduces the complexity to linear. Our goal is to make Transformers efficient on tabular datasets, especially for long sequences. As such, flash attention is not suitable for our objective.
>
> We believe that our paper makes sufficient technical contributions, demonstrates strong practical results, and holds significant potential for accelerating even broader tasks such as computer vision. Please refer to the `general response` for a summarized overview of our technical contributions. We are more than happy to help clarify any remaining confusion or address further concerns.
> > Q: Because switching to linear attention is such a straightforward process and does not require model retraining, it would be easy to transition all of the transformer-based methods described in the paper; this would be a fairer comparison, since the authors did not contribute a new learning algorithm. If this was done, I would expect most baselines to improve substantially, including TabPFN.
>
> Thank you for this thoughtful question. We respectfully clarify that switching to linear attention does *require model retraining*, and it is not universally applicable to all transformer-based models. We detail the reasons below.
>
> * **Performance Impact Without Retraining**: Directly replacing softmax attention with linear attention without retraining leads to suboptimal performance. Based on your feedback, we conducted experiments on 57 datasets with sizes ≤1250 (as used in Table 2 of the paper). The table below compares the performance of TabFlex (our method with retraining) to TabPFN, where the softmax function is replaced with a linear function without retraining.
>
>     | Dataset | #Classes | #Features | #Instances | Without Re-Training | With Re-Training (Ours) |
>     | --- | --- | --- | --- | --- | --- |
>     | Australian | 1 | 14 | 690 | 0.4550 | **0.9266** |
>     | Analcatdata_boxing1 | 1 | 3 | 120 | 0.3994 | **0.5544** |
>     | Heart-h | 1 | 13 | 294 | 0.5161 | **0.8885** |
>     | Breast-cancer | 1 | 9 | 286 | 0.6232 | **0.7102** |
>     | Iris | 3 | 4 | 150 | 0.1677 | **0.9987** |
>     | ... | ... | ... | ... | ... | ... |
>     | Average |  2.3684 | 18.7544 | 486.4737  | 0.4901 | **0.8583** |
>
>     As shown, retraining is essential to achieve the expected performance gains, and without retraining, the performance significantly degrades. This demonstrates that transitioning to linear attention cannot be considered a "straightforward process" for transformer-based methods.
>
> * **Transitioning All Transformer-Based Methods Is Highly Non-Trivial**: Transitioning all transformer-based methods to alternative architectures is not straightforward due to computational complexity and, in some cases, infeasibility. For example, TabLLM `(Hegselmann et al., 2023)` leverages the T0 model and employs few-shot learning. Replacing the softmax operation in the T0 model with a linear alternative significantly degrades performance, as demonstrated above. Moreover, pretraining a new T0 model using a linear attention mechanism is currently infeasible due to resource constraints.
>
>
> * **Focus of Our Work**: Our objective is to enhance the speed and performance of Transformer models. To achieve this, we focused our efforts on TabPFN, which stands out as the fastest Transformer-based method but is constrained to small datasets. Other Transformer models, in contrast, are significantly slower than TabPFN, as illustrated in Table 1 and 2 in our paper. Therefore, applying linear attention to other Transformer models is orthogonal to the goal of our paper.

---

> ### Author Response · Authors · 2024-11-26
>
> > Q: Given that the choice of linear attention method is a main technical contribution, it would have been very helpful to see comparisons to more baselines than just a SSM; there are many variants of linear attention available in PyTorch [https://pytorch.org/blog/flexattention/].
>
> Thank you for the excellent suggestion! While not all the methods fall strictly under the category of linear attention, we agree that testing on a broader range of model architectures provides a more comprehensive demonstration of our method's performance. To this end, we implemented the following two approaches:
>
> * Sliding Window Attention `(Beltagy et al., 2020)`
> * Bounded Attention (Soft-Capping) `(Team et al., 2024)`
>
> We argue that these approaches are not directly comparable to linear attention. Sliding window attention processes only a limited number of preceding tokens, which restricts its ability to leverage all training samples in our scenario. On the other hand, bounded attention does not effectively reduce computational complexity.
>
> To validate our approach, we conducted experiments on three datasets with extensive samples. Due to feasibility constraints, we applied soft-capping to include only 3,000 training samples, consistent with the TabPFN approach in `(McElfresh et al., 2023)`. The experimental results, which underscore these observations, are summarized in the table below.
>
> | Method     | #Class | #Features | #Instances | Sliding Window | Soft-Capping | Linear (Ours) |
> |------------|--------|-----------|------------|----------------|--------------|---------------|
> | Poker-Hand | 10     | 10        | 1025009    | 0.48           | 0.68         | **0.84**          |
> | Airlines   | 2      | 7         | 539383     | 0.48           | 0.62         | **0.64**          |
> | Higgs      | 2      | 28        | 98050      | 0.39           | 0.70          | **0.76**        |
>
> > Q: (Paraphrased) Argument 'In contrast, the causal model’s accuracy declines' is not sufficiently precise.
>
> Thank you for identifying this inaccuracy in the original description. Based on your feedback. We will revise it to: 'In contrast, the causal model's accuracy improves with limited training samples but deteriorates as more samples are added, highlighting poor sample utilization when ample data is available.'
>
> ***
> *References:*
> * `Ahn et al., 2023`. Transformers learn to implement preconditioned gradient descent for in-context learning. NeurIPS 2023.
> * `Hegselmann et al., 2023`. TabLLM: Few-shot Classification of Tabular Data with Large Language Models. AISTATS 2023.
> * `Beltagy et al., 2020`. Longformer: The Long-Document Transformer. ArXiv 2020.
> * `Team et al., 2024`. Gemma 2: Improving Open Language Models at a Practical Size. ArXiv 2024.
> * `McElfresh et al., 2023`. When do neural nets outperform boosted trees on tabular data? NeurIPS 2023 Datasets and Benchmarks.
>
> **Final Note:**  Thank you for reading our paper. We believe most of the concerns may have arisen from confusion. We will incorporate the discussion above into our final version. We hope we have clarified all your concerns. If so, we would highly appreciate it if you could *increase the score* and support our paper.

---

> > ### Comment · Reviewer_VkbL · 2024-11-26
> >
> > I thank the authors for a thoughtful and thorough rebuttal. Because some of my concerns have been addressed and clarified, I have raised my score from a 3 to a 5.

---

> > > ### Author Response · Authors · 2024-11-27
> > >
> > > Thank you for taking the time to review our response and for reconsidering your rating—it truly means a lot to us. If you have any additional suggestions or specific concerns preventing you from fully supporting the acceptance of our paper, we would be more than happy to address them. We will be uploading an updated PDF soon and will keep you informed of any further updates. Thank you again for your engagement with our work!

---

### Official Review · Reviewer_EvnT · 2024-11-05

**Soundness:** 4
**Presentation:** 4
**Contribution:** 2
**Rating:** 5
**Confidence:** 4

**Summary:**

This paper works on deep learning for tabular data, specifically using Transformers. The authors argued that existing Transformer-based models are "slow" either in training (need to re-train or fine-tune for each dataset) or inference (need to take the whole training set as input to provide the context). The authors pointed out that the standard attention module is the main cause (especially for long context in in-context learning), and proposed to replace it with linear attention. The authors conducted extensive experiments to show the superior performance of linear attention in tabular data --- it achieves significant speedup without sacrificing accuracy --- making Transformers with linear attention a suitable and preferable option for dealing with tabular data in an in-context learning fashion.

**Strengths:**

S1. The paper is very well-written and well-motivated, with adequate background and clear technical details.

S2. The paper conducts extensive experiments and analyses to justify the strength of linear attention for tabular data.

S3. The proposed idea is simple and widely applicable --- in my opinion, this is definitely a strength rather than a weakness. I could imagine if this paper is accepted (or just on arXiv), it will attract lots of attention; TabFlex will become a must-compare baseline.

I very much enjoy reading the paper.

**Weaknesses:**

That said, this paper has some critical weaknesses that the authors should try to address. I may miss some details; some weaknesses may have been addressed in the manuscript already. If so, please kindly point me to the sections and lines.

W1. First, I do not consider the analysis of Mamba as a contribution. I humbly do not see the feasibility of using a sequence model to solve in-context learning for tabular data. Thus, the poor performance by Mamba, to me, is expected. In my humble opinion, in-context learning for tabular data is a smart nearest neighbor (or non-parametric) algorithm. By default, a nearest-neighbor method should not rely on a sequence, unless there is an inherent sequence in the data. For example, if the authors show that ordering the training data based on their distance to the test data would improve Mambda's performance, then I think it will be an interesting finding and insight.

W2. The main weakness of the paper is the lack of insights. Yes, I see the extensive experiments and superior performance (time and accuracy). The reason for the efficiency/latency improvement is obvious and intuitive. However, I do not see a clear reason why linear attention would keep the same accuracy (or not sacrifice much) as standard attention. I'm not very familiar with the comparison between standard attention and linear attention in other contexts (e.g., LLMs) but I suppose that linear attention is not as good as standard attention (in terms of accuracy or reasoning capability) in many use cases; otherwise, it should have been the go-to module. It would be great if the authors could provide some more discussions and insights (not just quantitative empirical results) into why linear attention is a good option (beyond the latency aspect) in tabular data.

W3. While it may have been discussed in prior work, I would like to see a bit more background or justification as to why in-context learning is reasonable for solving tabular data. What does the Transformer really learn? Given that linear attention is inspired by inner products after learnable projections, it would be great for the authors to link in-context learning for tabular data to kernel-based methods.
FYI, I'm a supporter of using in-context learning to solve tabular data, but just hope that the paper can provide the readers with more insights beyond showing good performance.

W4. The conditional model selection looks quite naive. It just trains three models and uses a hard-coded rule to decide which model to use. What are the criteria to make the separation of the three models? What will be the accuracy of the three models given a fixed dataset? (I may miss some analyses; if so, please kindly point me to the corresponding lines or sections.)

===

I'm open to increasing my ratings based on how the authors address my concerns/weaknesses/questions. I see clear strengths of the paper. However, the current manuscript lacks critical insights, making it an excellent workshop paper but not a main conference paper yet.

**Questions:**

Besides the weaknesses above, I have a few more questions that I would like the authors to discuss OR compare.

Q1. Linear attention is particularly advantageous (in terms of time) in long-context scenarios. As traditional ML methods like SVM and nearest neighbors already show that the prediction of the test data can be reliably made based on a subset of training data, do we really need the whole training dataset as the context? For example, can we find a "core set" of training data to be the context? Or, can we order the training data by their distances to the test sample, and only use the closest 100 samples as the context? Will this really degrade the accuracy?

Q2. In section 3 background (Lines 153 - 157), all the test data are input to the model at once. Since the training samples do not attend to the test samples and the test samples do not attend to each other, inputting all the test samples is equivalent to inputting each of them one by one, but faster. In real-time inference, test data usually come one by one, not in a batch. Will such a speed-up still hold? (The answer to this question won't change my rating, but I'm just curious.)

---

> ### Author Response · Authors · 2024-11-26
>
> We would like to extend our special thanks to Reviewer EvnT for their *very constructive* feedbacks. Below, we address the concerns raised.
>
> ***
>
> > Q:  (Paraphrased) The authors should justify why linear attention maintains accuracy comparable to standard attention, beyond latency benefits, especially since standard attention dominates in accuracy-driven applications. Additionally, they should clarify why in-context learning is suitable for tabular data.
>
>
> This is an *excellent* question. Based on your comment, we will include the following discussion in our final version:
>
> **1. When does linear attention underperform standard attention?**
> `Han et al. (2024)` identify poor local modeling as a primary reason linear attention underperforms standard attention. Many real-world tasks, including tabular data analysis, rely heavily on capturing local dependencies for strong performance. For instance, in tabular data, nearby features often exhibit strong correlations that are critical for accurate predictions.
>
> However, in our approach, this limitation is mitigated because we represent each sample with a single token, effectively bypassing the challenges associated with poor local modeling.
>
> **2. When does linear attention work well?**
>    - **Theory:** Linear attention's capability for in-context learning has been theoretically proved. `Ahn et al. (2023)` propose that linear attention can effectively perform in-context learning by simulating gradient descent, providing theoretical support for the strong performance of our method.
>    - **Experiment:** A straightforward conjecture is that linear attention may resemble Mamba in its behavior, underperforming softmax attention in complicated tasks. The original paper did not directly address this, as we altered the architecture to support additional features and classes.
>
>        In our new experiments, we isolate the effect of the linear activation function in attention while keeping all other architectural components identical. To ensure feasibility for both TabPFN (softmax) and TabFlex-L100 (linear), we focus on tasks with <3,000 samples and <100 dimensions, considering low-dimensional datasets as simple and high-dimensional datasets as complex. Our results show that the performance gap between softmax and linear attention remains minimal across both categories, providing strong evidence that linear attention is a viable alternative for in-context learning.
>
>
>         |         | Dataset       | #Classes | #Features | #Instances | Softmax | Linear |
>         |---------|---------------|----------|-----------|------------|---------|--------|
>         |         |               |          |           |            |         |        |
>         | Simple  | balance-scale | 3        | 4         | 625        | 1.00    | 1.00   |
>         |         | credit-g      | 2        | 20        | 1000       | 0.77    | 0.76   |
>         |         |               |          |           |            |         |        |
>         | Complex | mfeat-fourier | 10       | 76        | 2000       | 0.98    | 0.96   |
>         |         | mfeat-zernike | 10       | 47        | 2000       | 0.98    | 0.96   |
>         |         |               |          |           |            |         |        |
>
>
> > Q: (Paraphrased) It would be great explore links between linear attention's inner-product-based mechanism and kernel methods to provide deeper theoretical insights.
>
> Thank you for the insightful question. In our paper, we adopt ELU+1 as the feature map due to its simplicity and strong practical performance, as demonstrated by `Katharopoulos et al. (2020)`. Building on your comments, we conducted experiments to compare the identity and ELU+1 kernel methods. Since replacing the kernel requires retraining, which typically takes several days, the results for the ReLU kernel are still in progress, and we will share them once available.
>
> The table below shows that ELU+1 significantly outperforms the identity feature map ,with only a small increase in runtime. This aligns with findings by` Arora et al. (2024)`, who observed that the choice of feature map plays a critical role in the memory-recall tradeoff. Simpler maps, such as ReLU and PosELU (ELU+1), often occupy the Pareto frontier of performance versus complexity.
>
> | Feature Map | tokyo1 | iris  | autoHorse | auto_price | tae   | vinne | xd6   | Relative Inference Time |
> |-------------|--------|-------|-----------|------------|-------|-------|-------|---|
> | Identity    | 0.968  | 0.994 | 0.966     | 0.991      | 0.655 | 0.918 | 0.865 | 1.0 $\times$ |
> | Elu + 1 (Ours)       | **0.979**  | **0.998** | **0.987**     | **0.995**      | **0.689** | **0.920** | **0.987** | 1.2 $\times$|

---

> > ### Author Response · Authors · 2024-11-26
> >
> > > Q: What are the criteria to make the separation of the three models? What will be the accuracy of the three models given a fixed dataset? (I may miss some analyses; if so, please kindly point me to the corresponding lines or sections.)
> >
> > The separation criteria are based on the architecture design and training paradigm. For example, TabFlex-S100 is trained on datasets with around 1,000 samples, and we observed that it performs well in practice on datasets with small sample sizes. The thresholds we chose are empirically determined to yield good performance.
> >
> > It's worth noting that the final performance is not particularly sensitive to these thresholds—small adjustments to the thresholds do not significantly impact the results. To provide more clarity, we present the performance of the three models across six datasets: two small and simple datasets, two large low-dimensional datasets, and two large high-dimensional datasets.
> >
> > |                          | Dataset        | Metric   | TabFlex-S100 | TabFlex-L100 | TabFlex-H1K | TabPFN |
> > |--------------------------|----------------|----------|--------------|--------------|-------------|--------------------|
> > |                          |                |          |              |              |             |                    |
> > |                          |                |          |              |              |             |                    |
> > | Simple                   | credit-g       | Accuracy | **0.82**         | 0.79         | 0.75        | 0.79               |
> > |                          |                | Time (s) | 0.13         | 0.13         | 0.13        | 0.23               |
> > |                          |                |          |              |              |             |                    |
> > |                          | diabet         | AUC      | **0.78**         | 0.77         | **0.78**        | **0.78**               |
> > |                          |                | Time (s) | 0.08         | 0.10         | 0.09        | 0.15               |
> > |                          |                |          |              |              |             |                    |
> > |                          |                |          |              |              |             |                    |
> > | Low-Dimensional & Large  | bank-marketing | AUC      | 0.89         | **0.90**         | 0.89        | 0.89               |
> > |                          |                | Time (s) | 0.25         | 2.43         | 1.67        | 1.75               |
> > |                          |                |          |              |              |             |                    |
> > |                          | elevators      | AUC      | 0.94         | **0.95**        | 0.94        | 0.94               |
> > |                          |                | Time (s) | 0.22         | 0.7          | 0.7         | 1.11               |
> > |                          |                |          |              |              |             |                    |
> > |                          |                |          |              |              |             |                    |
> > | High-dimensional & Large | nomao          | AUC      | 0.83         | 0.75         | **0.99**        | 0.86               |
> > |                          |                | Time (s) | 0.86         | 4.71         | 4.63        | 1.95               |
> > |                          |                |          |              |              |             |                    |
> > |                          | SpeedDating    | AUC      | 0.69         | 0.59         | **0.83**       | 0.66               |
> > |                          |                | Time (s) | 0.89         | 1.63         | 1.71        | 2.86               |
> > |                          |                |          |              |              |             |                    |
> >
> > Based on the table, we observe that for simple tasks, all models exhibit similar performance. However, TabFlex-S100 stands out for its efficiency, making it the preferred choice for simple tasks. For low-dimensional, large datasets, TabFlex-L100 demonstrates superior performance, while for high-dimensional, large datasets, TabFlex-H1K achieves the best results. This is primarily due to the increased difficulty in training with high-dimensional data, which necessitates selecting models that are better suited to different dimensionalities and dataset sizes.

---

> ### Author Response · Authors · 2024-11-26
>
> > Q: In section 3 background (Lines 153 - 157), all the test data are input to the model at once. Since the training samples do not attend to the test samples and the test samples do not attend to each other, inputting all the test samples is equivalent to inputting each of them one by one, but faster. In real-time inference, test data usually come one by one, not in a batch. Will such a speed-up still hold?
>
> Thank you for this interesting question. The speedup will still hold even if test data arrive one by one. This is because the training samples are fixed, and they do not attend to the test samples. As a result, all intermediate outputs of the training samples within the model remain unchanged, regardless of how the test data are processed.
>
> > Q:  Linear attention is particularly advantageous (in terms of time) in long-context scenarios. As traditional ML methods like SVM and nearest neighbors already show that the prediction of the test data can be reliably made based on a subset of training data, do we really need the whole training dataset as the context? For example, can we find a "core set" of training data to be the context? Or, can we order the training data by their distances to the test sample, and only use the closest 100 samples as the context? Will this really degrade the accuracy?
>
> This is an excellent suggestion. However, there are significant challenges in using nearest neighbors to select a "core set" for large datasets, primarily due to high computational overhead caused by two factors:
>
> * **Inability to use batch inference**: Since the in-context samples vary for each test instance, we need to recompute the attention outputs for every test sample individually. Our experiments demonstrate that without batch inference, inference times can increase by **1000×** or more in practice. For example, with 1000 test samples, we would typically classify all of them in a single forward pass. However, in this scenario, 1000 separate forward passes are required.
>
> * **Additional time complexity from sample selection**: Identifying and selecting the nearest samples introduces an extra computational burden, further impacting efficiency.
>
> Despite the high time complexity, we appreciate the reviewer’s insightful comment regarding the impact of sample selection on in-context learning, as it indeed represents an intriguing avenue for future research. In response, we conducted additional experiments on three datasets where TabPFN with standard random sample selection showed subpar performance. To enhance efficiency, we used TabPFN with 1000 sample selections based on distance (rather than 3000) and reported results based on 100 test samples.
>
> These experiments align with the reviewer’s expectations, demonstrating that a more strategic ordering of samples can significantly boost performance, albeit at the cost of reduced efficiency.
>
> | Dataset       | #Classes | #Features | #Instances | TabPFN (Random) | TabPFN (KNN) | TabFlex |
> |---------------|----------|-----------|------------|----------------|-----------------|---------|
> | SpeedDating   | 2        | 120       | 8378       | 0.55           | 0.73            | 0.85    |
> | Bioresponse   | 2        | 1776      | 3751       | 0.50           | 0.51            | 0.75    |
> | nomao         | 2        | 118       | 34465      | 0.76           | 0.99            | 0.99    |
>
> Nevertheless, your valuable feedback has inspired us to explore global sample selection, where important samples are selected from the entire dataset and reused across all test samples. This approach is commonly used in active learning and managing subpopulation shifts. However, typical sample selection techniques for active learning and subpopulation shifts usually require training a model first to identify important samples near the decision boundary, which conflicts with our goal of maintaining efficiency.
>
> To address this issue, we implement a simplified sample selection method. Specifically, we perform clustering on the dataset and then sample from different clusters to enhance diversity—a commonly recognized strategy to improve machine learning performance. In this experiment, we perform K-means on the training dataset with $k=10$, and then select 300 samples from each, resulting in total 3000 training samples. The results are presented below. We note that the results of TabPFN after this simple sample selection method is almost unchanged.
>
> | Dataset       | #Classes | #Features | #Instances | TabPFN (Random) | TabPFN (10-Means) | TabFlex |
> |---------------|----------|-----------|------------|----------------|-----------------|---------|
> | airlines   | 2        | 7     | 539383       | 0.63           | 0.63            | 0.64    |
> | poker-hand   | 10        | 10     | 1025009       | 0.72           | 0.71            | 0.84    |

---

> ### Author Response · Authors · 2024-11-26
>
> > Q: I humbly do not see the feasibility of using a sequence model to solve in-context learning for tabular data. Thus, the poor performance by Mamba, to me, is expected. In my humble opinion, in-context learning for tabular data is a smart nearest neighbor (or non-parametric) algorithm. By default, a nearest-neighbor method should not rely on a sequence, unless there is an inherent sequence in the data. For example, if the authors show that ordering the training data based on their distance to the test data would improve Mambda's performance, then I think it will be an interesting finding and insight.
>
>
> Thank you for this insightful question. To address the concern and clarify any confusion regarding the analysis of Mamba, we will incorporate the following discussion into the final version of our paper.
>
> * **Sequence models for in-context learning**: In fact, the use of sequence models for in-context learning has gained *significant* traction in recent research. For example, LIFT `(Dinh et al., 2022)` leverages the GPT-3 model for tabular classification, successfully utilizing dataset context. This approach has shown promise in diverse domains such as predictive chemistry and the physics of metamaterials.
>
> * **Mamba’s in-context learning capabilities and limitations**: While Mamba has been demonstrated to effectively solve linear regression tasks, it struggles with more complex scenarios, such as datasets generated by decision trees `(Park et al., 2024)`. Although the reasons for this limitation were not explicitly discussed in `Park et al. (2024)`, our work provides valuable insights by suggesting that causality is a contributing factor that degrades performance. our work provides valuable insights into these challenges by showin its causality can downgrade the performance. However, we also think additional factors may also be at play.
>
> * **Impact of Data Ordering on Performance**: Similar to sample selection, data ordering also incurs a significant computational overhead. Nonetheless, exploring the impact of data ordering on performance remains an interesting direction. Based on your feedback, we plan to incorporate the following experiments in the final version of our paper. Specifically, we will consider three data ordering strategies: (i) ordering samples in increasing distance to the test sample, (ii) ordering samples in decreasing distance, and (iii) random ordering.
>
>
>
> ***
>
> *References:*
> * `Han et al., 2024`. Bridging the divide: Reconsidering softmax and linear attention. NeurIPS 2024.
> * `Ahn et al., 2023`. Transformers learn to implement preconditioned gradient descent for in-context learning. NeurIPS 2023.
> * `Dinh et al., 2022`. LIFT: Language-interfaced fine-tuning for non-language machine learning tasks. NeurIPS 2022.
> * `Park et al., 2024`. Can mamba learn how to learn? a comparative study on in-context learning tasks. ICML 2024.
> * `Arora et al., 2024`. Simple linear attention language models balance the recall-throughput tradeoff. ArXiv 2024.
> * `Katharopoulos et al., 2020`. Transformers are rnns: Fast autoregressive transformers with linear attention. ICML 2020.
>
>
>
> **Final Note:** Many thanks for your detailed and insightful feedback. We recognize that the new discussion prompted by your comments has significantly strengthened our paper. In addition to the experiments detailed above, please check more new exciting results in general response. We also sincerely appreciate your kind note expressing a willingness to increase the scores if the concerns are addressed. We hope our response addresses most of your concerns, and we would be deeply grateful if you could help champion our paper.

---

> > ### Author Response · Authors · 2024-11-27
> >
> > Dear reviewer,
> >
> > We have updated our PDF file and conducted additional experiments based on your feedback. We greatly appreciate your time and effort, as we understand you may have a busy schedule. We hope our updates address all your concerns, and if they do, we would be grateful if you could consider increasing your score. Your support means a lot to us.
> >
> > Thanks in advance,
> > Authors

---

> > > ### Comment · Reviewer_EvnT · 2024-12-03
> > > **Response by the reviewer**
> > >
> > > Dear authors,
> > >
> > > Thank you for the detailed feedback. I do not have further questions for now. I will discuss the paper with other reviewers and accordingly adjust the score.

---

### Official Review · Reviewer_rEsm · 2024-11-06

**Soundness:** 3
**Presentation:** 3
**Contribution:** 3
**Rating:** 6
**Confidence:** 3

**Summary:**

The paper proposes a new method to process tabular data which can handle significantly more data and number of classes. Extensive experiments show that the proposed method achieves strong performances while being much faster compared to methods with similar performances.

**Strengths:**

1. The paper studies the problem of scaling up the application of attention-based methods to hundreds of classes with millions of data points. This significantly improves the applicability of similar methods on large datasets.
2. The paper studies various tradeoffs of introducing linear attentions and architectural alternatives where the authors shared lots of insightful findings.
3. The end performance is strong, as shown in Table 1 and 2, the method is able to achieve top 3 performances with little performance degradation compared to  other SOTA methods which require significant more time.
4. The paper is well written and the experiments include comprehensive number of baselines and datasets.

**Weaknesses:**

1. Looks like there is some formatting issue such as Figure 4.
2. Given the large amount of data presented in the tables, it's better to mark the top 3 with underline or bold so that we don't need to go over the large amount of data to find them out.
3. The use of different specialized models are good, are there any ways such as implementing a router instead of hard coding them.

**Questions:**

1. in Algorithm 1, how does random projection affect the final performances?
2. this brings another question, if we can randomly project the features to 1K, can we project them selectively? What's the benefit of scaling up with linear attention than just perform feature selection / augmentation?
3. Besides simply using ICL, maybe we can explore data pre-processing to aid the ICL to achieve better results?

---

> ### Author Response · Authors · 2024-11-26
>
> We appreciate the reviewer’s encouraging feedback, especially for recognizing that our paper is well-written, our paper provides lots of insightful findings by various studies and comprehensive experiments, and our method is significantly applicable to large datasets and strong in performance.
>
> ***
>
> > Q: Besides simply using ICL, maybe we can explore data pre-processing to aid the ICL to achieve better results?
>
> Thank you for this insightful question. In response, we conducted additional experiments on three datasets where TabPFN with standard random sample selection showed subpar performance. To enhance efficiency, we used TabPFN with 1000 sample selections based on distance (rather than 3000) and reported results based on 100 test samples. We observe that conducting sample selection can significantly improve the performance of ICL.
>
> | Dataset       | #Classes | #Features | #Instances | TabPFN (Random) | TabPFN (1000-NN) | TabFlex |
> |---------------|----------|-----------|------------|----------------|-----------------|---------|
> | SpeedDating   | 2        | 120       | 8378       | 0.55           | 0.73            | 0.85    |
> | Bioresponse   | 2        | 1776      | 3751       | 0.50           | 0.51            | 0.75    |
> | nomao         | 2        | 118       | 34465      | 0.76           | 0.99            | 0.99    |
>
> However, there are significant challenges in using this method for large datasets, primarily due to high computational overhead caused by two factors:
>
> * **Inability to use batch inference**: Since the in-context samples vary for each test instance, we need to recompute the attention outputs for every test sample individually. Our experiments demonstrate that without batch inference, inference times can increase by **1000×** or more in practice. For example, with 1000 test samples, our method classify all of them in a single forward pass. However, in this scenario, 1000 separate forward passes are required.
>
> * **Additional time complexity from sample selection**: Identifying and selecting the nearest samples introduces an extra computational burden, further impacting efficiency.
>
> While this method highlights promising performance improvements, it also opens avenues for future work, such as developing an optimized sample selection approach and addressing the incompatibility with batch inference.
>
> > Q: (paraphrased) Regarding Algorithm 1, what is the impact of random projection on the final performance? Additionally, if we can reduce the features through random projection to 1,000 dimensions, is there an option to make this process selective rather than random?
>
> Thanks for this insightful question. Based on your feedback, we conducted a new set of experiments on MNIST and Fashion-MNIST to evaluate the impact of different selection methods: random selection ("Random Select"), random projection ("Random Projection"), selecting the first n samples ("First Select"), and selecting features based on mutual information ("Mutual Information"). The result table below indicate that random projection achieves strong performance while maintaining reasonable computational overhead.
>
> |               |          | First Select | Random Select | Random Projection | Mutual Information |
> |---------------|----------|-----------------------------|------------------------------|----------------------------------|-----------------------------------|
> |               |          |                             |                              |                                  |                                   |
> | MNIST         | AUC      | .791                        | .943                         | **.948**                         | .924                              |
> |               | Time (s) | .706 (1×)                   | .785 (1.11×)                 | .771 (1.09×)                     | .853 (1.21×)                      |
> |               |          |                             |                              |                                  |                                   |
> | Fashion-MNIST | AUC      | .917                        | .974                         | **.979**                         | .964                              |
> |               | Time (s) | .746 (1×)                   | .785 (1.05×)                 | .810 (1.09×)                     | .886 (1.19×)                      |

---

> ### Author Response · Authors · 2024-11-26
>
> > Q: Moreover, what are the advantages of scaling with linear attention compared to simply performing feature selection or augmentation?
>
> Thank you for the insightful question. Linear attention is not in conflict with feature selection—they actually address different challenges. Linear attention is primarily aimed at handling *larger numbers of samples*, while feature selection focuses on managing datasets with *high dimensionality*.
>
> A natural follow-up question is whether sample selection could be a viable alternative to linear attention. Our answer is no, because selecting samples individually for each test instance prevents efficient *batch inference*, which is crucial for maintaining fast inference times. Since the in-context samples vary for each test instance, we would need to recompute the attention outputs for every test sample individually. Our experiments show that without batch inference, the inference time can increase by over **1000×** in practice. For example, with 1000 test samples, we would normally classify them all in a single forward pass. However, in this scenario, we would need 1000 separate forward passes, making it highly inefficient.
>
> > Q: The use of different specialized models are good, are there any ways such as implementing a router instead of hard coding them.
>
> Thank you for this insightful suggestion. The current separation criteria are based on the architecture design and training paradigm. Implementing a router for automatic model selection is an interesting direction for future work. We have added this to our future work section and will upload updated PDF soon.
>
> > Q: Given the large amount of data presented in the tables, it's better to mark the top 3 with underline or bold so that we don't need to go over the large amount of data to find them out.
>
> Thank you for the suggestion. In Tables 1 and 2, the rows (each representing a method) are ordered by performance, which is why we chose not to use boldface. However, based on your feedback, we have applied boldface to Table 4 and will upload the updated PDF shortly.
>
>
> > Q: Looks like there is some formatting issue such as Figure 4.
>
> Thanks for pointing it out. Fixed and will upload the updated PDF soon.
>
> ***
>
> **Final Note:** Thank you for your valuable comments. We are grateful to hear that you found our paper provides many insightful findings, and that our method is significantly applicable to large datasets and demonstrates strong performance. In addition to the experiments detailed above, please check more new exciting results in general response. If our responses have resolved your concerns, we kindly request you to consider increasing your score and support the acceptance of our paper.

---

> > ### Author Response · Authors · 2024-11-27
> >
> > Dear reviewer,
> >
> > We have updated our PDF file and conducted additional experiments based on your feedback. We greatly appreciate your time and effort, as we understand you may have a busy schedule. We hope our updates address all your concerns, and if they do, we would be grateful if you could consider increasing your score. Your support means a lot to us.
> >
> > Thanks in advance,
> > Authors

---

> > ### Comment · Reviewer_rEsm · 2024-12-02
> >
> > Thank the authors for the responses. I maintain positive about this work and recommend acceptance. Good luck!

---

> > > ### Author Response · Authors · 2024-12-02
> > >
> > > Thank you for taking the time to review our response and for your kind words of support. We greatly appreciate your recommendation for acceptance, and it truly means a lot to us!

---

### Official Review · Reviewer_zg6o · 2024-11-06

**Soundness:** 3
**Presentation:** 3
**Contribution:** 2
**Rating:** 5
**Confidence:** 4

**Summary:**

This paper focuses on extending the applicability of in-context learning (ICL) to a larger scale of datasets without sacrificing performance and latency. Specifically, the author expands on the scalability of TABPFN, where they first provide an investigation of the state-space models (SSMs) and linear attention. The empirical findings show that compared to SSMs, linear attention is better for maintaining comparable performance while improving computational efficiency. The author also conducts extensive comparisons with other methods across a diverse range of datasets, showing that the proposed method, i.e., TABFLEX, has robust performance with impressive computational efficiency.

**Strengths:**

(1) Extending the applicability of in-context learning to larger datasets for tabular classification is crucial for many real-world applications.

(2) The identified scalability limitation of TabPFN is relevant; the corresponding attention choice, i.e., the non-causal linear attention, can also achieve comparable performance with TabPFN with a large speedup.

**Weaknesses:**

(1) The empirical findings in 4.1 seem to be restricted to TabPFN, and it is hard to see whether they can be applied to a broader range of in-context learning designs. The author may want to show that they are general to other frameworks of in-context learning.

(2) The technical novelty is limited. The author leverages the empirical study of two kinds of attention mechanisms to identify the "optimal" candidate for TABPFN, as stated in lines 291-292. Although the empirical study is relevant, it does not bring novel technical contributions to the community as the author does not propose a new technical design to even further improve linear attention. Moreover, given that all the design choice is tailored to TABPFN, it is hard to convince the reviewer that the present paper brings sufficient contributions to the community. If the author wants to justify that the linear attention is exactly optimal and there is no other choice better than it, then the reviewer may suggest that the author provide rigorous theoretical analysis to justify their choice.

Minor:

(1) There is a formatting issue in Fig. 4.

**Questions:**

Please refer to the Weaknesses section.

**Details Of Ethics Concerns:**

N/A.

---

> ### Author Response · Authors · 2024-11-26
>
> We thank the reviewer for acknowledging that our paper identifies important limitations of prior work and that our method is crucial for real-world applications.
>
>
> ***
>
>
> > Q: Limited technical contribution and insights.
>
> Thank you very much for your critical feedback. We clarify our work and added several new ablation studies based on your suggestions. Below, we summarize the key insights from both the original and new experiments. We believe these insights, along with the strong performance of our method, make it both practical and valuable. All the points discussed here will be incorporated into the final version.
>
> * **`(NEW EXPERIMENT)` Insight 1: A New Avenue for Accelerating Image Classification**
>
>     Our method extends beyond tabular data and shows potential for significantly speeding up image classification in computer vision tasks.
>
>     To demonstrate this, we compared TabFlex to a standard MLP—a basic neural network approach. Note that we did not perform extensive hyperparameter tuning for MLP, as tuning would add significant computational overhead. We tested both a two-layer and a three-layer MLP, each with 10 hidden neurons, using a fixed learning rate of 0.001 across 70 epochs. Our results indicate that TabFlex achieves approximately 30× speedup over MLP.
>
>     The key takeaway is that TabFlex opens an entirely new avenue for accelerating image classification, which was impossible with previous methods like TabPFN, due to their limitation to low-dimensional, small datasets. Although MNIST is a relatively simple dataset, our experiments provide a foundation for future exploration in this direction. This work offers a comprehensive examination of how to make such advancements possible.
>
>     |               | Two-Layer MLP      |                | Three-Layer MLP      |                | TabFlex (Ours)       |          |
>     |---------------|--------------------|----------------|----------------------|----------------|----------------------|----------|
>     |               | AUC                | Time (s)       | AUC                  | Time (s)       | AUC                  | Time (s) |
>     | MNIST         | .924               | 23.547 (30.5x) | .959                 | 23.06 (29.9x)  | .948                 | .771     |
>     | Fashion-MNIST | .793               | 23.340 (28.8x) | .853                 | 23.604 (29.1x) | .979                 | .810     |
>
>
> * **`(NEW EXPERIMENTS)` Insight 2: TabFlex: A Close-to-Optimal Solution for Large-Scale High-Dimensional Classification.**
>
>     The previous insight emphasizes the potential of TabFlex for high-dimensional, large datasets. Here, we provide an ablation study showing that TabFlex is nearly optimal both theoretically and empirically.
>
>     * **Evidence 2.1: Linear Attention as the Close-to-Optimal Choice for Scaling to Large Datasets.**
>
>         Clearly, there are other potential methods such as data subsampling and other alternatives for attentions.
>         * Finding 2.1.1: Data Selection Increases Computational Overheads Significantly.
>
>             We conducted additional experiments on three datasets where TabPFN with standard random sample selection underperformed. To enhance efficiency, we employed TabPFN with 1000 nearest-neighbor (KNN) sample selections (instead of 3000) and evaluated results based on 100 test samples. Our findings show that sample selection significantly improves ICL performance.
>
>             | Dataset       | #Classes | #Features | #Instances | TabPFN (Random Sample Selection) | TabPFN (KNN Sample Selection) | TabFlex |
>             |---------------|----------|-----------|------------|----------------|-----------------|---------|
>             | SpeedDating   | 2        | 120       | 8378       | 0.55           | 0.73            | 0.85    |
>             | Bioresponse   | 2        | 1776      | 3751       | 0.50           | 0.51            | 0.75    |
>             | nomao         | 2        | 118       | 34465      | 0.76           | 0.99            | 0.99    |
>
>             However, there are significant challenges in using this method for large datasets, primarily due to high computational overhead caused by two factors:
>
>             1. Inability to use batch inference: Since the in-context samples vary for each test instance, we need to recompute the attention outputs for every test sample individually. Our experiments demonstrate that without batch inference, inference times can increase by **1000×** or more in practice. For example, with 1000 test samples, our method classify all of them in a single forward pass. However, in this scenario, 1000 separate forward passes are required.
>
>             2. Additional time complexity from sample selection: Identifying and selecting the nearest samples introduces an extra computational burden, further impacting efficiency.

---

> ### Author Response · Authors · 2024-11-26
>
> * [Indent]
>     * [Indent]
>         * Finding 2.1.2: Superior Performance of Linear Attention Compared to Other Sub-Quadratic Complexity Models
>
>             In addition to the big categories of all linear RNN variant models we studied in the paper, we consider another mechanism enjoys linear complexity: sliding window attention `(Beltagy et al., 2020)`. We show that TabFlex enjoys much better performance.
>             | Method     | #Class | #Features | #Instances | Sliding Window | Linear (Ours) |
>             |------------|--------|-----------|------------|----------------|--------------|
>             | Poker-Hand | 10     | 10        | 1025009    | 0.48         | **0.84**          |
>             | Airlines   | 2      | 7         | 539383     | 0.48       | **0.64**          |
>             | Higgs      | 2      | 28        | 98050      | 0.39           | **0.76**        |
>
>     * **Evidence 2.2: Random Projection for Efficient Scaling in High-Dimensional Settings in Addition to a Widder Model.**
>
>         We conducted a new set of experiments on MNIST and Fashion-MNIST to evaluate the impact of different selection methods: random selection ("Random Select"), random projection ("Random Projection"), selecting the first n samples ("First Select"), and selecting features based on mutual information ("Mutual Information"). The result table below indicate that random projection (used by TabFlex) achieves strong performance while maintaining reasonable computational overhead.
>
>         |               |          | First Select | Random Select | Random Projection | Mutual Information |
>         |---------------|----------|-----------------------------|------------------------------|----------------------------------|-----------------------------------|
>         |               |          |                             |                              |                                  |                                   |
>         | MNIST         | AUC      | .791                        | .943                         | **.948**                         | .924                              |
>         |               | Time (s) | .706 (1×)                   | .785 (1.11×)                 | .771 (1.09×)                     | .853 (1.21×)                      |
>         |               |          |                             |                              |                                  |                                   |
>         | Fashion-MNIST | AUC      | .917                        | .974                         | **.979**                         | .964                              |
>         |               | Time (s) | .746 (1×)                   | .785 (1.05×)                 | .810 (1.09×)                     | .886 (1.19×)                      |
>     * **Evidence 2.3: Linear Attention is HBM-Efficient.**
>
>         Through theoretical analysis, we prove that the existing implementation of linear attention is inherently nearly optimal, and making it I/O-aware results in only marginal improvements, as demonstrated in Theorem 2 and Table 4.
>
>     * **Evidence 2.4: Linear Attention is an Effective In-Context Learner.**
>         While linear attention may underperform in certain language modeling tasks, it has been shown to be effective for in-context learning tasks `(Ahn et al., 2023)`. Our experiments confirm its efficacy, particularly in tabular data settings involving long sequences.
>
> * **Insight 3: Strong Performance and Speedup in Tabular Classification Tasks.**
>     This is demonstrate by our extensive experiments in our paper.
>
>
> > Q: The author may want to show that they are general to other frameworks of in-context learning.
>
> Thank you for this insightful question. To clarify, our primary goal is to establish a fundamental stepping stone towards accelerating classification for high-dimensional datasets. This focus is distinct from improving general in-context learning techniques, making our work somewhat orthogonal to applying our method to standard in-context learning.
>
> Furthermore, applying our approach to existing in-context learning frameworks is not practical due to significant differences in methodology. Standard in-context learning typically leverages very large pre-trained models like GPT-3, whereas our approach involves training a model from scratch with a completely new architecture. Pretraining such a large model would not only be impractical given resource constraints but also contradict our goal of enhancing classification speed and efficiency.
>
> > Q: There is a formatting issue in Fig. 4.
>
> Thanks for pointing it out. Fixed and will upload the updated PDF soon.

---

> ### Author Response · Authors · 2024-11-26
>
> ***
>
> *References:*
>
> * `Ahn et al., 2023`. Transformers learn to implement preconditioned gradient descent for in-context learning. NeurIPS 2023.
> * `Beltagy et al., 2020`. Longformer: The Long-Document Transformer. ArXiv 2020.
>
>
>
> **Final Note:** We are grateful of your great suggestion. Based on your comment, we summarize the great potential of our work, and add many ablation studies to show the optimality of our method. If our responses have resolved your concerns, we kindly request you to consider *increasing your score* and support the acceptance of our paper.

---

> ### Author Response · Authors · 2024-11-27
>
> Dear reviewer,
>
> We have updated our PDF file and conducted additional experiments based on your feedback. We greatly appreciate your time and effort, as we understand you may have a busy schedule. We hope our updates address all your concerns, and if they do, we would be grateful if you could consider increasing your score. Your support means a lot to us.
>
> Thanks in advance,
>
> Authors

---

### Author Response · Authors · 2024-11-26
**To AC and All Reviewers**

We would like to thank all reviewers for their comments and helpful feedback. We are particularly encouraged that the reviewers have found that (i) our method is crucial for real-world applications `(R-zg6o)`, strong in performance `(R-rEsm)`, widely `(R-EvnT)` and significantly `(R-rEsm)` applicable to large datasets, broader in range and utility `(R-VkbL)`, and simple `(R-EvnT)`; (ii) our paper identifies relevant limitations of prior work `(R-zg6o)`, provides lots of insightful findings through various studies `(R-rEsm)`, includes many `(R-VkbL)` experiments and analyses `(R-EvnT)`, and is comprehensive `(R-rEsm)` and extensive `(R-EvnT)`; (iii) we release code for reproducibility `(R-VkbL)`; and (iv) our paper has adequate background and clear technical details `(R-EvnT)`, and is very well written `(R-rEsm, R-EvnT)`, well-motivated `(R-EvnT)`, and clear `(R-VkbL)`.


In response to the reviewers' feedback, we have carefully addressed each concern and supplemented our work with additional experimental results where necessary. All the new discussion will be incorporated in the final version. To provide clarity, we would like to emphasize two key points:

* Clarification: Our paper does *not* utilize flash attention and *requires* re-training.
* Technical Contribution: We have further refined our summary of the technical contribution and added new experiments to strengthen our findings, summarized as below.


# `(NEW EXPERIMENT)` Insight 1: A New Avenue for Accelerating Image Classification

Our method extends beyond tabular data and shows potential for significantly speeding up image classification in computer vision tasks.

To demonstrate this, we compared TabFlex to a standard MLP—a basic neural network approach. Note that we did not perform extensive hyperparameter tuning for MLP, as tuning would add significant computational overhead. We tested both a two-layer and a three-layer MLP, each with 10 hidden neurons, using a fixed learning rate of 0.001 across 70 epochs. Our results indicate that TabFlex achieves approximately 30× speedup over MLP.

The key takeaway is that TabFlex opens an entirely new avenue for accelerating image classification, which was impossible with previous methods like TabPFN, due to their limitation to low-dimensional, small datasets. Although MNIST is a relatively simple dataset, our experiments provide a foundation for future exploration in this direction. This work offers a comprehensive examination of how to make such advancements possible.

|               | Two-Layer MLP      |                | Three-Layer MLP      |                | TabFlex (Ours)       |          |
|---------------|--------------------|----------------|----------------------|----------------|----------------------|----------|
|               | AUC                | Time (s)       | AUC                  | Time (s)       | AUC                  | Time (s) |
| MNIST         | .924               | 23.547 (30.5x) | .959                 | 23.06 (29.9x)  | .948                 | .771     |
| Fashion-MNIST | .793               | 23.340 (28.8x) | .853                 | 23.604 (29.1x) | .979                 | .810     |

# `(MAJOR, NEW EXPERIMENTS)` Insight 2: TabFlex: A Close-to-Optimal Solution for Large-Scale High-Dimensional Classification.

The previous insight emphasizes the potential of TabFlex for high-dimensional, large datasets. Here, we provide an ablation study showing that TabFlex is nearly optimal both theoretically and empirically.

## Evidence 2.1: Linear Attention as the Close-to-Optimal Choice for Scaling to Large Datasets.

Clearly, there are other potential methods such as data subsampling and other alternatives for attentions.

### Finding 2.1.1: Data Selection Increases Computational Overheads Significantly.

We conducted additional experiments on three datasets where TabPFN with standard random sample selection underperformed. To enhance efficiency, we employed TabPFN with 1000 nearest-neighbor (KNN) sample selections (instead of 3000) and evaluated results based on 100 test samples. Our findings show that sample selection significantly improves ICL performance.

| Dataset       | #Classes | #Features | #Instances | TabPFN (Random Sample Selection) | TabPFN (KNN Sample Selection) | TabFlex |
|---------------|----------|-----------|------------|----------------|-----------------|---------|
| SpeedDating   | 2        | 120       | 8378       | 0.55           | 0.73            | 0.85    |
| Bioresponse   | 2        | 1776      | 3751       | 0.50           | 0.51            | 0.75    |
| nomao         | 2        | 118       | 34465      | 0.76           | 0.99            | 0.99    |

However, there are significant challenges in using this method for large datasets, primarily due to high computational overhead caused by two factors:

---

> ### Author Response · Authors · 2024-11-26
>
> 1. Inability to use batch inference: Since the in-context samples vary for each test instance, we need to recompute the attention outputs for every test sample individually. Our experiments demonstrate that without batch inference, inference times can increase by 1000× or more in practice. For example, with 1000 test samples, our method classify all of them in a single forward pass. However, in this scenario, 1000 separate forward passes are required.
>
> 2. Additional time complexity from sample selection: Identifying and selecting the nearest samples introduces an extra computational burden, further impacting efficiency.
>
> ### Finding 2.1.2: Superior Performance of Linear Attention Compared to Other Sub-Quadratic Complexity Models
>
> In addition to the big categories of all linear RNN variant models we studied in the paper, we consider another mechanism enjoys linear complexity: sliding window attention `(Beltagy et al., 2020)`. We show that TabFlex enjoys much better performance.
> | Method     | #Class | #Features | #Instances | Sliding Window | Linear (Ours) |
> |------------|--------|-----------|------------|----------------|--------------|
> | Poker-Hand | 10     | 10        | 1025009    | 0.48         | **0.84**          |
> | Airlines   | 2      | 7         | 539383     | 0.48       | **0.64**          |
> | Higgs      | 2      | 28        | 98050      | 0.39           | **0.76**        |
>
> ## Evidence 2.2: Random Projection for Efficient Scaling in High-Dimensional Settings in Addition to a Widder Model.
>
> We conducted a new set of experiments on MNIST and Fashion-MNIST to evaluate the impact of different selection methods: random selection ("Random Select"), random projection ("Random Projection"), selecting the first n samples ("First Select"), and selecting features based on mutual information ("Mutual Information"). The result table below indicate that random projection (used by TabFlex) achieves strong performance while maintaining reasonable computational overhead.
>
> |               |          | First Select | Random Select | Random Projection | Mutual Information |
> |---------------|----------|-----------------------------|------------------------------|----------------------------------|-----------------------------------|
> |               |          |                             |                              |                                  |                                   |
> | MNIST         | AUC      | .791                        | .943                         | **.948**                         | .924                              |
> |               | Time (s) | .706 (1×)                   | .785 (1.11×)                 | .771 (1.09×)                     | .853 (1.21×)                      |
> |               |          |                             |                              |                                  |                                   |
> | Fashion-MNIST | AUC      | .917                        | .974                         | **.979**                         | .964                              |
> |               | Time (s) | .746 (1×)                   | .785 (1.05×)                 | .810 (1.09×)                     | .886 (1.19×)                      |
> ## Evidence 2.3: Linear Attention is HBM-Efficient.
>
> Through theoretical analysis, we prove that the existing implementation of linear attention is inherently nearly optimal, and making it I/O-aware results in only marginal improvements, as demonstrated in Theorem 2 and Table 4.
>
> ## Evidence 2.4: Linear Attention is an Effective In-Context Learner.
>
> While linear attention may underperform in certain language modeling tasks, it has been shown to be effective for in-context learning tasks `(Ahn et al., 2023)`. Our experiments confirm its efficacy, particularly in tabular data settings involving long sequences.
>
> # `(MAJOR)` Insight 3: Strong Performance and Speedup in Tabular Classification Tasks.
>
> This is demonstrate by our extensive experiments in our paper.
>
> *References:*
>
> * `Ahn et al., 2023`. Transformers learn to implement preconditioned gradient descent for in-context learning. NeurIPS 2023.
> * `Beltagy et al., 2020`. Longformer: The Long-Document Transformer. ArXiv 2020.

---

### Meta-Review · Area_Chair_5bms · 2024-12-22

**Metareview:**

The paper is borderline and received mixed reviews post-rebuttal. In sum, the paper explores efficient alternatives to self-attention based Transformers for in-context learning on tabular classification, and settles on linear-attention as the one that's scalable to millions in sequence length. The finding is also generalized to image classification during rebuttal. However, none of the reviewers are enthusiastic about the work, despite the numerous authors' efforts on including new results, iterating on the draft, and responding to reviewers which the AC appreciates. There are also remaining to-dos as the authors have promised, e.g., clarifying on not using FlashAttention, which increases the uncertainty on additional contents that won't have gone through reviews if accepted. Weighing all the factors, the AC decides the paper would benefit from another cycle. To further improve the quality, studies of other efficient alternatives beyond Mamba (e.g., RWKV) could be a possibility.

**Additional Comments On Reviewer Discussion:**

Please see the metareview for the factors considered. Overall, the concerns and potentials for further improvements outweigh the benefits of accepting the paper in the current cycle.

---

### Decision · Program_Chairs · 2025-01-22

Reject